# Distribution and fluxes of marine particles in the South China Sea continental slope: implications for carbon export

Shujin Guo[1, 2†], Mingliang Zhu[1, 2†], Wenlong Xu[3], Shan Zheng[1, 2], Sumei Liu[4], Ying Wu[5], Juan Du[1, 2], Chenhao Zhao[1], Xiaoxia Sun[1, 2, 6]

[1]Jiaozhou Bay National Marine Ecosystem Research Station, Institute of Oceanology, Chinese Academy of Sciences, Qingdao 266071, China

[2]Laboratory for Marine Ecology and Environmental Science, Qingdao Marine Science and Technology Center, Qingdao 266237, China

[3]Ocean College, Jiangsu University of Science and Technology, Zhenjiang 212100, China

[4]Frontiers Science Center for Deep Ocean Multispheres and Earth System, and Key Laboratory of Marine Chemistry Theory and Technology, Ministry of Education, Ocean University of China, Qingdao 266100, China

[5]State Key Laboratory of Estuarine and Coastal Research, Faculty of Earth Sciences, East China Normal University, Shanghai 201100, China

[6]University of Chinese Academy of Sciences, Beijing 100049, China

*Correspondence to:* Xiaoxia Sun (xsun@qdio.ac.cn)

**Abstract.** Marine particles are key vectors in the ocean's biological carbon pump, yet their distribution, size structure, contributions to particulate organic carbon (POC) flux, and the mechanisms controlling these processes remain poorly understood in marginal seas. In this study, we investigated the spatial distribution and carbon flux of marine particles along the continental slope of the South China Sea (SCS), using *in situ* imaging data collected by an Underwater Vision Profiler during a June 2015 cruise. We also examined how these particle-related processes respond to mesoscale eddy activity. Particle abundance and volume concentration (PVC) ranged from 0 to 783 particles $L^{-1}$ (mean $\pm$ SD: 68 $\pm$ 69 particles $L^{-1}$) and from 0 to 6.7 $mm^3$ $L^{-1}$ (mean $\pm$ SD: 0.3 $\pm$ 0.4 $mm^3$ $L^{-1}$), respectively. Small particles, which were defined as those with an equivalent spherical diameter less than 500 μm, overwhelmingly dominated in terms of abundance, accounting for more than 97% of total counts. However, in terms of PVC, large particles contributed a greater share, averaging 61% $\pm$ 12%. PVC was significantly higher in the epipelagic layer (mean $\pm$ SD: 0.4 $\pm$ 0.7 $mm^3$ $L^{-1}$) than in the mesopelagic layer (mean $\pm$ SD: 0.2 $\pm$ 0.1 $mm^3$ $L^{-1}$, $p < 0.01$), indicating enhanced particle production in surface waters. Under the influence of mesoscale eddies, distinct differences in particle characteristics and carbon export were observed. Cyclonic eddies enhanced particle concentrations and favored the formation of large particles, while anticyclonic eddies were associated with a higher proportion of small particles. These patterns were linked to eddy-induced changes in nutrient availability and phytoplankton production. Consequently, POC fluxes in cyclonic eddy–influenced regions were consistently higher than those in anticyclonic regions throughout the water column, with POC fluxes reaching over twice the magnitude observed in anticyclonic eddy regions. This suggests that mesoscale eddies can influence carbon export by altering both the concentration and size composition of marine particles. Our study clarifies the distribution and size structure of marine particles along the SCS slope and highlights the importance of mesoscale physical processes in regulating particle–mediated carbon export, thereby enhancing our understanding of carbon cycling processes in dynamic marginal seas.

---

† These authors contributed equally to this work.

**Key words:** marine particles, Underwater Vision Profiler, particle distribution, size structure, mesoscale eddies, carbon export, South China Sea, continental slope

## 1 Introduction

Marine particles are critical components of the oceanic carbon cycle, serving as vehicles for transporting organic carbon from the surface ocean to the deep sea via the biological pump (Boyd et al., 2019; Siegel et al., 2022). These particles, which include a diverse array of forms such as micron-scale phytoplankton cells,

submillimeter detrital fragments, millimeter-scale aggregates, and zooplankton fecal pellet, play an essential role in global carbon dynamics (Turner, 2015). The abundance and distribution of marine particles are regulated by a combination of abiotic and biotic factors, such as primary productivity, particle aggregation and fragmentation, as well as hydrodynamic conditions all interact to shape particle fields in the ocean (Forest et al., 2012; Kiko et al., 2022). Due to the widespread vertical settling behavior of particles in the ocean, their abundance and size not

only influence the efficiency of the carbon export but also govern the biogeochemical pathways through which carbon is transformed, remineralized, or permanently stored in the ocean's interior (Kiko et al., 2022). Therefore, understanding the distribution, size structure, and associated carbon export of marine particles is critical for assessing the efficiency of the biological carbon pump.

Although marine particles play a crucial role in ocean biogeochemical cycles, their fragile nature poses

significant challenges for collection and analysis. Sediment traps have been widely employed to capture settling particles and quantify vertical fluxes (Honjo et al., 2008; Harms et al., 2021). While this method provides valuable insights, it lacks the spatial coverage and detailed particle size distribution data necessary for a mechanistic understanding of particle dynamics (Wang et al., 2024a). In recent years, advancements in *in situ* optical and imaging technologies have introduced an alternative approach for assessing particle distribution and

estimating fluxes (Picheral et al., 2010; Boss et al., 2015). The Underwater Vision Profiler (UVP) has emerged as a powerful tool, enabling high-resolution, *in situ* measurements of particle abundance and size distribution across a broad depth range (Picheral et al., 2010). When combined with image processing techniques for particle identification, the UVP facilitates comprehensive characterization of particle abundance, size, composition, and potential sources (Trudnowska et al., 2021, 2023), thereby enhancing our understanding of particle formation

and transformation processes. Beyond particle characterization, the UVP has been instrumental in estimating particle fluxes by leveraging empirical relationships between particle size and sinking flux (Guidi et al., 2008a; Iversen et al., 2010; Ramondenc et al., 2016). The high vertical resolution of UVP observations in marine particles, coupled with established size-dependent relationships for carbon content and sinking velocity (Kriest, 2002; Guidi et al., 2008a; Clements et al., 2023), provides a uniquely detailed perspective on the three-

dimensional distribution of particulate organic carbon (POC) flux in the ocean (Guidi et al., 2016; Kiko et al., 2017; Cram et al., 2018).

Mesoscale eddies, as ubiquitous and energetic features in the global ocean, have been extensively studied for their roles in regulating physical dynamics, biogeochemical processes, and ecosystem structures (McGillicuddy Jr et al., 2016). Numerous studies have demonstrated that eddies can significantly influence

nutrient transport, primary productivity, and biological community composition in the ocean (Stramma et al., 2013; Zhang et al., 2020; Barone et al., 2022). Cyclonic eddies generally induce upwelling that brings cold, nutrient-rich subsurface water into the euphotic zone, thereby stimulating new production and enhancing

phytoplankton biomass (Shih et al., 2020). This nutrient enrichment often favors the growth of large-sized phytoplankton, particularly diatoms, leading to a community shift toward a more productive, herbivore-based food web (Barlow et al., 2014; Wang et al., 2016). In contrast, anticyclonic eddies are typically characterized by downwelling and enhanced water column stratification, which suppress vertical nutrient supply, leading to reduced surface productivity and a food web dominated by smaller phytoplankton and microbial pathways (Peterson et al., 2005; Stramma et al., 2013). Despite these advances, our understanding of how mesoscale eddies influence the distribution and size structure of marine particles remains limited, particularly in marginal seas. Key questions remain unanswered: To what extent do mesoscale eddies regulate particle abundance and size composition? Do cyclonic and anticyclonic eddies exert distinct influence due to their contrasting physical and biogeochemical characteristics? And critically, how do these differences in particle dynamics translate into variability in vertical carbon export? The lack of observations and systematic analyses on these issues hampers accurate assessments on biological pump efficiency in the ocean. Therefore, elucidating the impacts of mesoscale eddies on particles characteristics, including abundance, size composition, and vertical transport potential, is essential for improving our understanding of oceanic carbon cycling.

The South China Sea (SCS) is one of the largest semi-enclosed marginal seas, characterized by a narrow and steep continental slope in its northern region (Zhang et al., 2020). This slope features a rapid bathymetric gradient, with depths increasing sharply from less than 200 m at the shelf break to over 1000 m. Mesoscale eddies are ubiquitous in the slope region of the SCS, where both cyclonic and anticyclonic eddies frequently occur due to the combined influence of monsoonal winds, Kuroshio intrusion, and complex topography (Wu and Chiang, 2007; Shih et al., 2020; Zhang et al., 2020). Typically spanning 100–200 km in diameter and persisting for weeks to months, these eddies exhibit pronounced seasonal variability (Wu and Chiang, 2007). Cyclonic eddies are more prevalent during the summer months, driven by enhanced upwelling and the southwesterly monsoon, whereas anticyclonic eddies are more frequently observed in winter and early spring, often formed as warm-core rings shed from the Kuroshio Current loop (Shih et al., 2020; Zhang et al., 2020). These eddies propagate predominantly westward or southwestward along the northern continental slope of the SCS, significantly modulating the vertical structure of temperature, salinity, and nutrient fields in this region (Guo et al., 2015). While the impacts of mesoscale eddies on physical circulation, nutrient dynamics, and phytoplankton community structures have been relatively well studied in this region (Wu and Chiang, 2007; Xiu and Chai, 2011; Wang et al., 2016; Xu et al., 2025), their influence on the distribution and size structure of marine particles, especially in the context of carbon export processes, remains insufficiently explored. This lack of direct observational constraints hinders our understanding of how marine particles regulate carbon export pathways and respond to mesoscale physical processes.

This study aims to fill the existing knowledge gaps by analyzing high-resolution UVP data collected along the continental slope of the SCS during a 2015 cruise, during which a mesoscale anticyclonic-cyclonic eddy pair was observed. Specifically, the objectives of this study are to (1) provide the first detailed characterization of the spatial distribution of marine particles along the SCS slope and identify the key environmental factors driving their variability, (2) analyze the size composition of particles and evaluate its implications for carbon cycling, and (3) investigate the mechanisms by which mesoscale eddies modulate particles dynamics and their associated POC fluxes. This study will elucidate the distribution patterns and controlling factors of marine particles along the SCS slope, and provide new insights into how eddy-induced physical variability regulates particle dynamics and their associated vertical transfer in the SCS.

## 2 Material and methods

### 2.1 Surveyed area and data acquisition

A research cruise was carried out in the SCS aboard the research vessel *Nanfeng* from June 13 to June 29, 2015. The study area and survey stations were shown in Fig. 1a. The survey stations can be divided into three transects (Transect 1, Transect 2, and Transect 3) (Fig. 1b), which extend across the slope of the SCS, with shallower sites at over 100 m and deeper sites exceeding 1000 m in depth (Table 1). Satellite data were used to analyze the spatial and temporal variations in the physical and dynamic characteristics of the study region. The Sea Surface Salinity (SSS) data consists of daily global, gap-free Level-4 (L4) analyses of SSS at a resolution of 1/8°. These analyses are generated through a multivariate optimal interpolation algorithm that combines SSS data from different satellite sources, including NASA's Soil Moisture Active Passive (SMAP) and ESA'S Soil Moisture Ocean Salinity (SMOS) satellites, along with *in situ* salinity measurements provided by the Copernicus Marine Environment Monitoring Service (Multi Observation Global Ocean Sea Surface Salinity and Sea Surface Density) (Buongiorno Nardelli et al., 2016; Sammartino et al., 2022). Eddies were identified as local minima (cyclonic eddies) and maxima (anticyclonic eddies) in sea level anomaly (SLA) using a composite altimetry product, which is based on a combination of remote sensing observations distributed by the Copernicus Marine Environment Monitoring Service (SEALEVEL_GLO_PHY_CLIMATE_L4_MY_008_057).

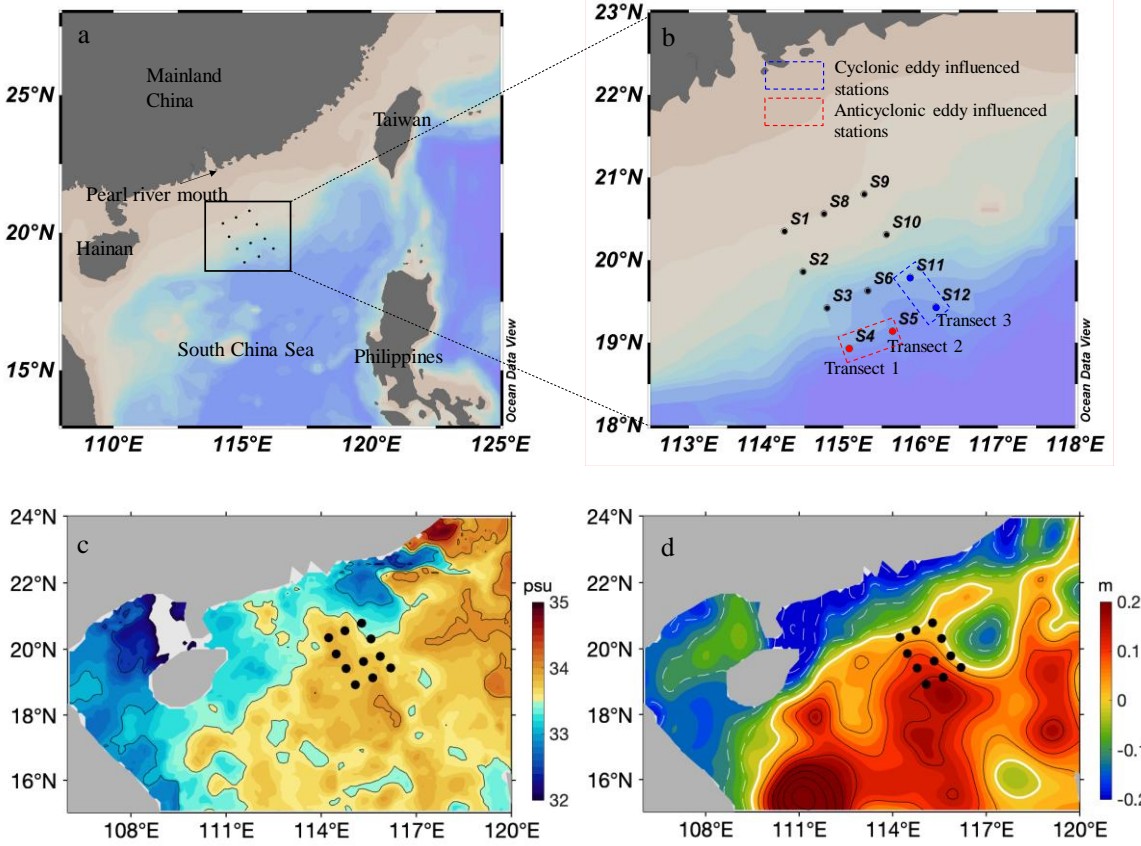

**Fig. 1 Study area and survey stations in the SCS continental slope. a: the study region; b: the enlarge view; c: sea surface salinity (SSS) at 1/8° of resolution; d: sea level anomalies (SLA). The dots indicate the locations of the sampling stations. Red dots indicate stations influenced by the anticyclonic eddy, and blue dots represent stations located at the periphery of the cyclonic eddy.**

**Table 1. Location and deployment times of UVP5-HD in the study area. UVP deployment depth refers to the maximum depth reached during each profile. Sampled volume represents the total volume imaged by the UVP5-HD during each deployment. Station 7 was skipped due to instrument malfunction, and therefore not included in this study.**

| Station | Longitude (ºE) | Latitude (ºN) | Date | Time (hh:mm, UTC + 8) | Bottom depth (m) | UVP5-HD deployment depth (m) | Total particles ($\times 10^4$ #) | Sampled volume ($\times 10^3$ L) |
|---------|---------------|---------------|------|------------------------|------------------|------------------------------|-----------------------------------|----------------------------------|
| S1 | 114.40 | 20.29 | 18-June-2015 | 02:40 p.m. | 123 | 110 | 101.74 | 5.99 |
| S2 | 114.47 | 19.87 | 17-June-2015 | 03:45 a.m. | 587 | 500 | 182.55 | 12.80 |
| S3 | 114.79 | 19.42 | 16-June-2015 | 01:35 p.m. | 1300 | 800 | 52.73 | 13.85 |
| S4 | 115.08 | 18.93 | 15-June-2015 | 11:45 p.m. | 3000 | 800 | 64.79 | 15.39 |
| S5 | 115.64 | 19.14 | 15-June-2015 | 07:30 a.m. | 2800 | 800 | 65.49 | 15.04 |
| S6 | 115.32 | 19.63 | 14-June-2015 | 03:25 p.m. | 2108 | 800 | 66.24 | 15.77 |
| S8 | 114.75 | 20.56 | 18-June-2015 | 08:50 p.m. | 120 | 100 | 95.83 | 6.47 |
| S9 | 115.27 | 20.80 | 19-June-2015 | 08:15 a.m. | 170 | 110 | 49.97 | 7.28 |
| S10 | 115.56 | 20.31 | 19-June-2015 | 07:30 p.m. | 560 | 500 | 69.98 | 5.84 |
| S11 | 115.86 | 19.78 | 20-June-2015 | 01:20 p.m. | 1556 | 800 | 131.91 | 15.25 |
| S12 | 116.20 | 19.43 | 21-June-2015 | 01:45 p.m. | 1971 | 800 | 51.39 | 5.64 |

Hydrographic measurements and water sampling were conducted using an SBE911 plus dual

conductivity-temperature-depth (CTD) sensor unit coupled with an SBE 32 Water Sampler (Seabird

Scientific, Bellevue, WA, USA). The CTD system recorded profiles of temperature (℃), salinity (psu),

and pressure (dbar) throughout the water column.

Nutrients and chlorophyll $a$ (Chl $a$) concentrations were determined from seawater samples

collected at multiple depths using a 10 L Niskin sampler (KC-Denmark Inc., Denmark) deployed

alongside a CTD. For nutrient analysis, seawater samples were filtered through a 0.45 μm pore-size

cellulose acetate membrane, and the filtrates were stored at temperatures below –20 ℃ until further

processing. In the laboratory, nutrient concentrations were measured using a Technicon AA3

autoanalyzer (Bran-Luebbe GmbH) following standard protocols (Liu et al., 2022). The detection limits

for nutrient analysis were 0.02 μmol L$^{-1}$ for $NO_3^-$, 0.01 μmol L$^{-1}$ for $NO_2^-$, 0.03 μmol L$^{-1}$ for $PO_4^{3-}$, and

0.05 μmol L$^{-1}$ for $SiO_3^{2-}$. Chl $a$ concentrations were determined using the fluorometric method

(Welschmeyer, 1994). Seawater samples (500 mL) were filtered through 0.7 μm Whatman GF/F filters,

and pigments were extracted in 90% acetone at 4 ℃ in the dark for 24 h. Fluorescence measurements

were then conducted using a Turner Designs fluorometer (Model 10).

**2.2 Particle measurement**

Particle size and abundance were measured using a high-resolution, high-frequency Underwater

Vision Profiler (UVP5-HD). The UVP5-HD was mounted downward-facing on the CTD Niskin-

rosette, and vertical deployments were conducted at a descent speed of 1 m/s. The UVP5-HD captured

images of illuminated particles within a known sampling volume of 1.053 L per frame. Particle size

was determined based on the number of pixels in the captured images, and the pixel-to-metric unit

conversion followed the standard calibration procedures described by Picheral et al. (2010), which

were established through laboratory experiments using natural particles in a seawater tank. Images

were recorded digitally at a rate of 12 frames per second and processed using custom developed image

analysis software from the Laboratoire d'Océanologie de Villefranche-sur-Mer, including ZooProcess

(v7.22) and PkID (v1.26). The equivalent spherical diameter (ESD) of each particle was calculated

under the assumption that the particle's projected shape was circular. Particle volume concentration

(PVC) was calculated by summing the estimated volumes of all particles detected within each depth

bin and normalizing by the corresponding sampled water volume. The volume of each particle was

estimated by assuming spherical geometry based on the ESD. The final PVC is reported in $mm^3$ $L^{-1}$.

Previous inter-calibration studies of UVP systems have shown that only the overlapping size

ranges among different studies are suitable for comparative particle profile analyses (Guidi et al.,

2008a; Picheral et al., 2010). For consistency with previous studies, we set the upper size limit for

particle flux calculations at 1.5 mm ESD (Guidi et al., 2007, 2008a; Stemmann et al., 2008;

Ramondenc et al., 2016). The lower size limit was set at 100 μm ESD to exclude signals potentially

caused by camera resolution constraints and background noise, which could not be reliably

distinguished as actual particles (Fender et al., 2019). For data visualization in this study, particles were

categorized following Kiko et al. (2022) into two groups: small particles (ESD < 0.50 mm) and large

particles (ESD ≥ 0.50 mm). This classification enables the examination of size-dependent vertical

distribution patterns, which can reflect underlying processes such as aggregation near the surface,

disaggregation at depth, and differential sinking behavior of particles of varying sizes.

To assess the influence of mesoscale eddies on the biological component of the particle field,

large particles were classified into living and non-living categories. In this context, living particles refer

specifically to large zooplankton, while all others were categorized as non-living particles. We

extracted and analyzed zooplankton data from the UVP5-HD imagery. Due to the resolution limitations

of the UVP5-HD, small particles could not be reliably imaged or morphologically classified in this

study. Vignette images of large particles were extracted from the UVP5-HD dataset and used for

zooplankton classification. The extracted images were uploaded to the EcoTaxa platform

(https://ecotaxa.obs-vlfr.fr), where a machine learning classifier (random forest algorithm) was applied

to assign objects into broad morphological categories, including zooplankton and non-living particles

(Picheral et al., 2017). All classified images were then visually checked and corrected by trained

analysts to ensure taxonomic accuracy. The abundance of zooplankton (ind. $L^{-1}$) was calculated by

dividing the number of individuals identified in each depth bin by the corresponding volume of water

imaged by the UVP5-HD.

**2.3 Estimation of POC export flux from particle size spectrum**

The POC export flux was estimated from particle size spectra using the method developed by

Guidi et al. (2008a, b). The particle size distribution (PSD) generally follows a power-law decrease

over the μm to mm size range (Guidi et al., 2009). This distribution, derived from UVP5-HD images, is

expressed as:

$$n(d) = \alpha d^{\beta} \quad (1)$$

where $d$ (mm) represents particles diameter, $n(d)$ is the particle size spectrum, $\alpha$ is normalization

constant and $\beta$ is the exponent characterizing the slope of the number spectrum after log

transformation. The particle size-based carbon flux approach assumes that the total carbon flux, ($F$)

(mg C m$^{-2}$ d$^{-1}$), corresponds to the integral of the flux spectrum over all particle sizes, from the smallest

($d_{min}$) to the largest ($d_{max}$) diameter:

$$F = \int_{d_{min}}^{d_{max}} n(d) \cdot m(d) \cdot w(d) dd \qquad (2)$$

where $n(d)$ is the particle number spectrum (particles m$^{-3}$ mm$^{-1}$), $m(d)$ (mg C) is the carbon mass of a

particle with diameter $d$, and $w(d)$ (m d$^{-1}$) is its sinking velocity.

The combined particle mass and settling velocity follow a power-law function of particle

diameter, based on empirical relationships derived from comparisons of PSDs obtained through

imaging systems and sediment trap mass flux estimates (Guidi et al., 2008a; Jouandet et al., 2011). This

relationship is expressed as: $m(d) \cdot w(d) = Ad^{B}$, where $A$ (mg C m d$^{-1}$) and $B$ are empirical constants. The

particle carbon flux can thus be approximated by discretizing Equation (2) into small logarithmic

diameter intervals (Guidi et al., 2009; Picheral et al., 2010):

$$F = \sum_{i=1}^{x} n_i Ad_i^{B} \Delta d_i \quad (3)$$

where $A = 12.5 \pm 3.40$ and $B = 3.81 \pm 0.70$, representing the best-fit parameters that minimized log-

transformed discrepancies between global oceanic sediment trap-derived carbon flux estimates and

UVP-derived particle abundance and PSDs (Guidi et al., 2008a). This approach has been widely

applied in various oceanic regions worldwide in recent years (Iversen et al., 2010; Ramondenc et al.,

2016; Fender et al., 2019; Clements et al., 2023; Wang et al., 2024a, b).

**2.4 Data analysis**

All particle size and abundance data obtained from the UVP5-HD were binned into 5 m vertical

intervals and subsequently processed to generate depth profiles of particle abundance (particles L$^{-1}$),

particle volume concentration (mm$^3$ L$^{-1}$), and POC flux (mg C m$^{-2}$ d$^{-1}$). To visualize along-slope

variability, section plots were created for each transect by interpolating these depth-resolved parameters

across stations. Statistical analyses were performed using SPSS 25.0. Two-tailed independent-sample $t$-

tests were conducted to assess significant differences in particle and flux parameters between different

depth layers and regions. Prior to applying *t*-tests, data normality and variance homogeneity were tested

using the Shapiro-Wilk and Levene's tests, respectively (González-Estrada and Cosmes, 2019). For data

not meeting parametric assumptions, non-parametric alternatives were used. Pearson correlation analysis

was applied to evaluate the relationships between POC flux and environmental variables. Significance

was defined at $p < 0.05$ unless otherwise stated. Hydrographic and particle data were visualized using

Ocean Data View 4 and Origin 2022.

**3 Results**

**3.1 Hydrographic conditions**

Based on the spatial distribution of SLA, two mesoscale eddies were present in the study area

during the survey period: one cyclonic eddy and one anticyclonic eddy (Fig. 1d). Stations S4 and S5,

located on the outer edge of Transects 1 and 2, respectively, were situated in a region where SLA

exceeded +0.1 m, indicating influence from an anticyclonic eddy. In contrast, a distinct cyclonic eddy

with SLA values below –0.1 m was present to the east of stations S11 and S12 in Transect 3. These two

stations were positioned along the periphery of the cyclonic eddy, where the SLA gradient was steep.

Vertically, the study area exhibits distinct characteristics of a tropical ocean, with relatively high sea

surface temperature (~30 ℃) and strong stratification in the upper 200 m (Fig. 2). A pronounced

thermocline is observed between 100 and 200 m, marking a sharp temperature gradient (Fig. 2a-c).

Salinity increases with depth, further reinforcing the stratification pattern (Fig. 2d-f).

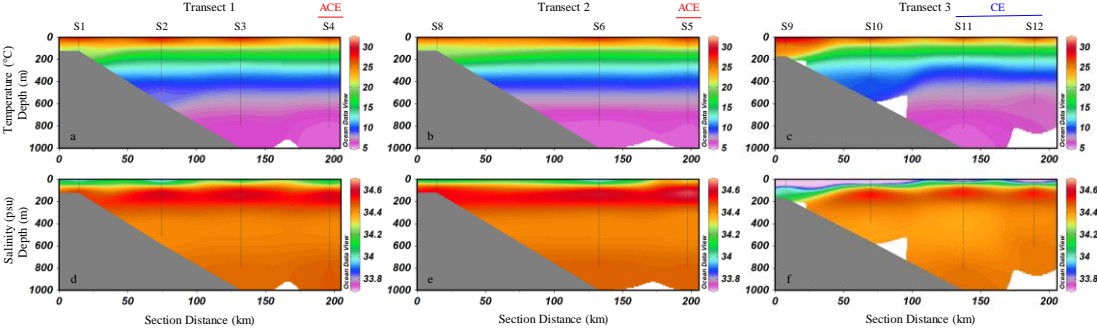

**Fig. 2. Vertical distribution of temperature (a-c, ℃) and salinity (d-f, psu) along the three transects in the**

**study area.**

The sectional distributions of nutrients and Chl *a* concentrations in the study area are shown in Fig. 3. Nutrient profiles exhibit typical oligotrophic features, with low concentrations of nitrate, phosphate and silicate in surface waters and a pronounced nutricline between 50 and 100 m. Chl *a* exhibited a pronounced deep chlorophyll maximum, especially along Transect 1 and 2 (Fig. 3j, k). Along Transect 3, the depth of the DCM became shallower, and Chl *a* concentrations were relatively high throughout the upper layer at nearshore stations (Fig. 3i). There were clear differences in the concentration of nutrients and Chl *a* between stations influenced by the anticyclonic eddy (S4, S5) and those influenced by the cyclonic eddy (S11, S12) (Supplementary Fig. S1). Nutrient concentrations at stations S11 and S12 were generally higher than those at stations S4 and S5 from 0 to 200 m (Supplementary Fig. S1a-c). Chl *a* concentrations were higher at stations S11 and S12 than at stations S4 and S5 in the upper 50 m layer (Supplementary Fig. S1d). With increasing depth, Chl *a* levels at S11 and S12 gradually declined, whereas those at S4 and S5 continued to rise, peaking at approximately 75 m before decreasing again.

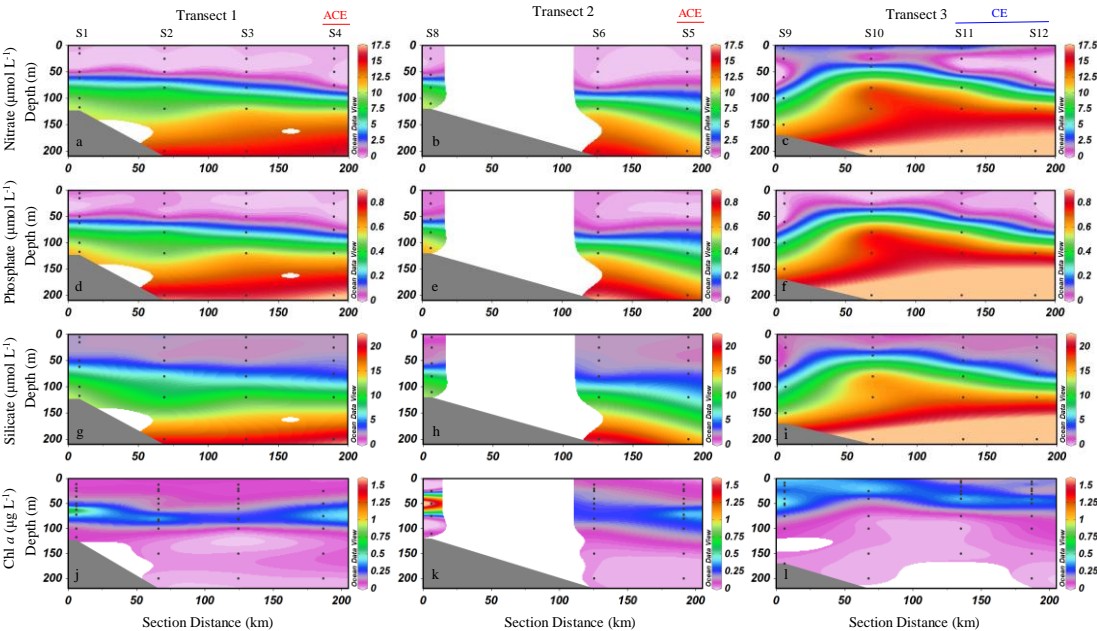

**Fig. 3. Sectional distribution of nutrient concentrations (nitrate, phosphate, and silicate) (μmol L⁻¹) and Chl *a* concentrations (μg L⁻¹) in the upper 200 m of the study area. a-c: nitrate (μmol L⁻¹); d-f: phosphate (μmol L⁻¹); g-i: silicate (μmol L⁻¹); j-l: Chl *a* (μg L⁻¹). ACE denotes stations influenced by the anticyclonic eddy, while CE denotes stations influenced by the cyclonic eddy.**

## 3.2 Particle concentration and size structure

Particle abundances ranged from 0 to 783 particles L$^{-1}$, with a mean of $68 \pm 68$ particles L$^{-1}$ (mean $\pm$ SD). The vertical profiles of particle abundance across the three transects were shown in Fig. 4. In Transect 1, high particle abundance was observed at station S1 and the bottom layer of station S2, with values reaching up to 700 particles L$^{-1}$ (Fig. 4a). In Transect 2, particle abundance gradually decreased from the inner shelf to the outer slope and from surface to bottom waters (Fig. 4b). In Transect 3, a distinct peak in particle abundance was observed at 379–390 m depth at the slope station S10 (Fig. 4c). Furthermore, at the outermost stations S11 and S12, elevated particle concentrations were observed in the upper 50 m of the water column, with peak values reaching up to 200 particles L$^{-1}$. This is notably higher than the maximum particle abundances recorded at stations S4 and S5, where values remained below 150 particles L$^{-1}$ (Fig. 4a, b). The distribution patterns of zooplankton (Supplementary Fig. S2) in the study area were generally similar with that of particle abundances. On transects 1 and 2, higher zooplankton abundances were observed at the nearshore stations, with a decreasing trend toward offshore stations (Supplementary Fig. S2a, b). In contrast, on transect 3, elevated abundances were recorded at the outer stations S11 and S12 (Supplementary Fig. S2c). Regarding particle size composition, small particles overwhelmingly dominated the total particle abundance, consistently accounting for more than 97% across all transects and throughout the water column (Fig. 4d-f). In contrast, large particles contributed only a minor fraction of the total abundance at all depths and locations (Fig. 4g-i).

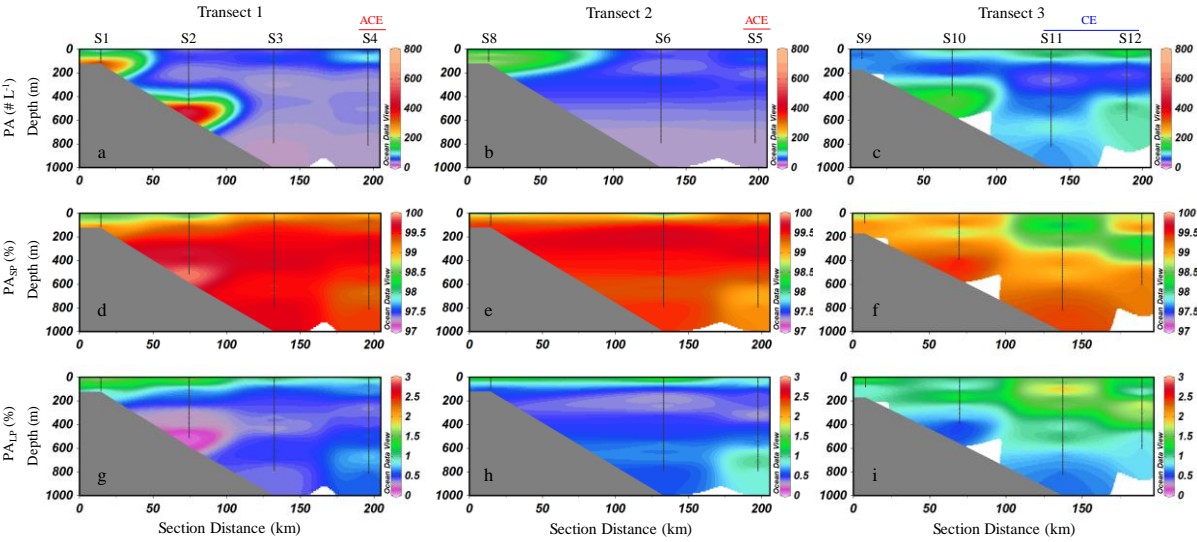

**Fig. 4 Vertical distribution of particles abundance (PA, # L$^{-1}$) and the relative proportion of small particles (PA$_{SP}$, ESD < 500 μm) and large particles (PA$_{LP}$, ESD ≥ 500 μm) to total abundance. PA: a-c; PA$_{SP}$%: d-f; PA$_{LP}$%: g-i. SP: small particles; LP: large particles. ACE represents stations influenced by anticyclonic eddies, while CE represents stations influenced by cyclonic eddies. Interpolated values between S6 and S8 are automatically generated by ODV and should be interpreted with caution.**

The distribution of PVC along the three transects, as well as the proportions of small and large particles contributing to PVC, are shown in Fig. 5. PVC ranged from 0 to 6.7 mm$^3$ L$^{-1}$, with a mean of $0.3 \pm 0.4$ mm$^3$ L$^{-1}$ (mean $\pm$ SD) in this study. Overall, high PVC values in the study area were primarily observed in the upper water column, while values decreased markedly as water depth increased (Fig. 5a-c). In waters shallower than 200 m, the mean PVC was $0.4 \pm 0.7$ mm$^3$ L$^{-1}$ (mean $\pm$ SD), which was significantly higher than that in waters deeper than 200 m, where the mean PVC was $0.2 \pm 0.1$ mm$^3$ L$^{-1}$ (mean $\pm$ SD) (t-test, $p < 0.01$). In Transect 1, PVC was highest

at the inner slope station S1 and progressively decreased toward the outer slope stations (Fig. 5a). A similar pattern was observed in Transect 2, where PVC was elevated at the inner slope station S8, followed by a gradual decline toward the outer slope (Fig. 5b). In Transect 3, PVC was relatively high in upper layers at the outermost stations S11 and S12, with concentrations decreasing with depth (Fig. 5c). For the whole study region, small particles contribute between 13% and 74% (mean ± SD: 39% ± 12%) of the total PVC (Fig. 5d-f), while large particles account for 26% to 87% (mean ± SD: 61% ± 12%) (Fig. 5g-i). For small particles, their contribution to PVC was generally around 40%–50% across most areas (Fig. 5d-f). However, notable variations were observed in certain regions. At station S2, a sharp increase occurred in the bottom layer, where the proportion exceeded 60% (Fig. 5d). In contrast, at stations S11 and S12, a marked decrease was observed in the upper to mid-depth layers, where the proportion dropped to 30% or even lower (Fig. 5f).

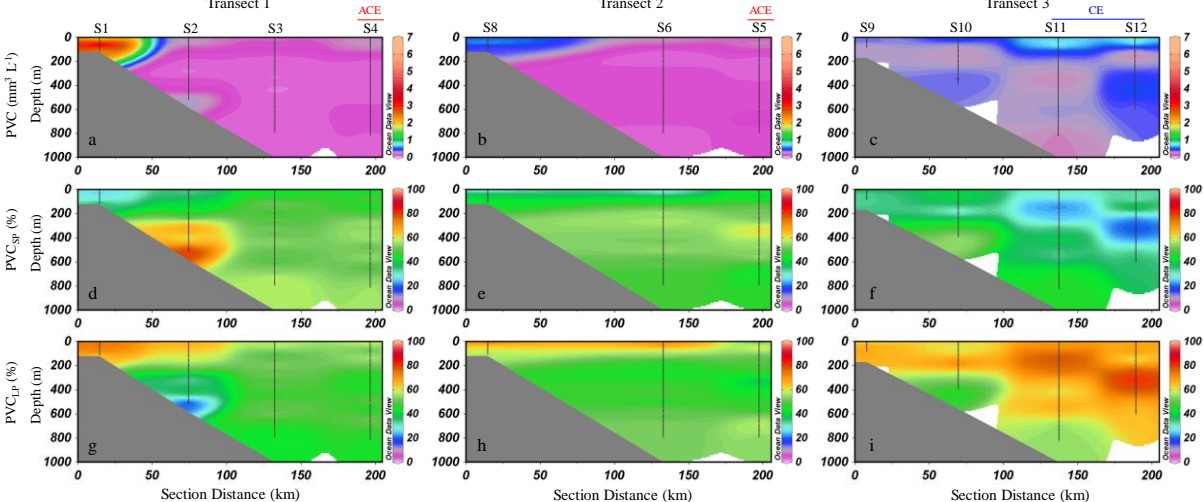

**Fig. 5. Distribution of particle volume concentration (PVC) (mm³ L⁻¹) along the three transects and the percentage contributions of small and large particles. PVC: a-c; percentage of small particles: d-f; percentage of large particles: g-i. SP: small particles; LP: large particles. ACE represents stations influenced by anticyclonic eddies, while CE represents stations influenced by cyclonic eddies. Interpolated values between S6 and S8 are automatically generated by ODV and should be interpreted with caution.**

Figure 6 shows the PVC values at different depths and the percentage contributions of small and large particles at stations influenced by anticyclonic eddies (S4, S5) and cyclonic eddies (S11, S12). Notable differences in PVC values and particle size composition can be observed between these two regions. At nearly all depth, PVC at S11 and S12 is higher than at S4 and S5 (Fig. 6a). At S4 and S5, small particles generally contribute more than 40% to PVC across most depths, except at 200 m at S4, whereas large particles remain below 60% (Fig. 6b, c). At S11 and S12, the proportion of small particles is mostly below 40%, especially above 500 m, with large particles contributing the majority of the volume (Fig. 6d, e). The proportion of large particles at S11 and S12 is higher than at S4 and S5.

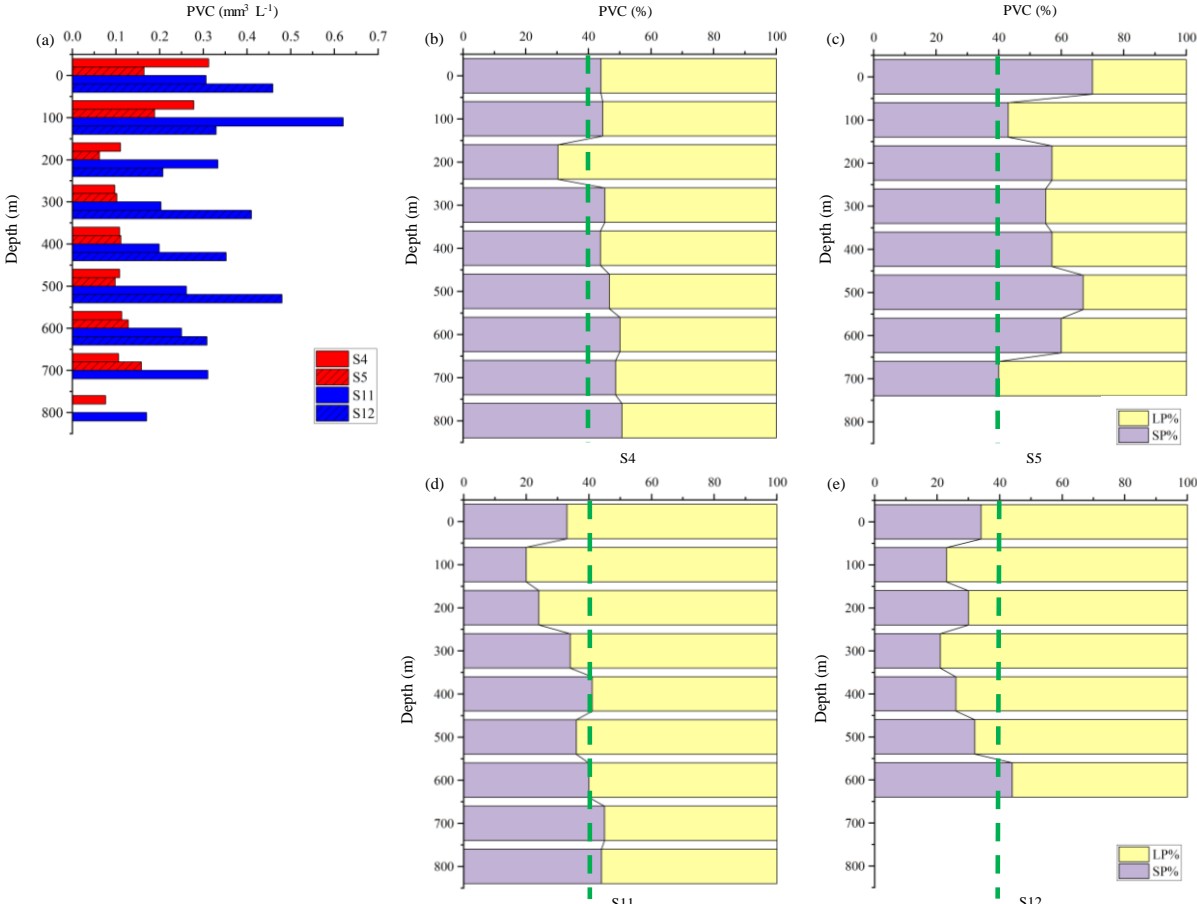

**Fig. 6. PVC (mm³ L⁻¹) at different depths at stations influenced by anticyclonic eddies (S4, S5) and cyclonic eddies (S11, S12) and the contributions of small particles and large particles. a: PVC at each layer; b-e: size composition of PVC at each station. SP: small particles; LP: large particles. The green dashed line represents a 40% reference threshold to visually distinguish the dominant size class. This threshold is not based on statistical testing but serves as a comparative guide.**

### 3.3 POC export flux and spatial variability

The POC export flux, derived from UVP-based particle size distributions, exhibited a wide range from 3.4 to 302.4 mg C m⁻² d⁻¹, with a mean value of 33.6 ± 34.9 mg C m⁻² d⁻¹ across the study area. Overall, high POC fluxes were primarily concentrated in the upper 200 m (Fig. 7). In Transect 1, the highest fluxes were concentrated at the nearshore station S1, particularly within the 50–70 m depth range, where values peaked at 302.0 mg C m⁻² d⁻¹ (Fig. 7a). In contrast, offshore stations along the same transect displayed significantly lower fluxes, generally below 40 mg C m⁻² d⁻¹. A similar nearshore-to-offshore gradient was observed in Transect 2, where the maximum flux (164.0 mg C m⁻² d⁻¹) was recorded at 40 m depth at station S8, gradually decreasing toward deeper slope stations (Fig. 7b). Transect 3 showed a distinct pattern, with elevated POC fluxes occurring at offshore stations S11 and S12 (Fig. 7c). At station S11, the highest flux (212 mg C m⁻² d⁻¹) was recorded at 60 m, while at station S12, the flux peaked at 210 mg C m⁻² d⁻¹ at 40 m. .

The contribution of small and large particles to POC flux across the study area was shown in Supplementary Fig. S3. Small particle-derived carbon flux accounted for 5% to 77% of the total POC flux, with an average

contribution of 24% ± 12%, while large particle-derived carbon flux ranged from 23% to 95%, averaging 76% ± 12%. The contribution of large particles to the POC flux was significantly higher than that of small particles ($t$-test, $p < 0.05$). Vertically, in waters shallower than 200 m, small particles contributed 19 ± 9% (mean ± SD) to the POC flux, while large particles accounted for 81 ± 9% (mean ± SD). In contrast, below 200 m, the contribution of small particles increased to 28 ± 12% (mean ± SD), with large particles contributing 72 ± 12% (mean ± SD). The contribution of small particles to POC flux below 200 m was significantly higher than that in the upper 200 m ($t$-test, $p < 0.05$).

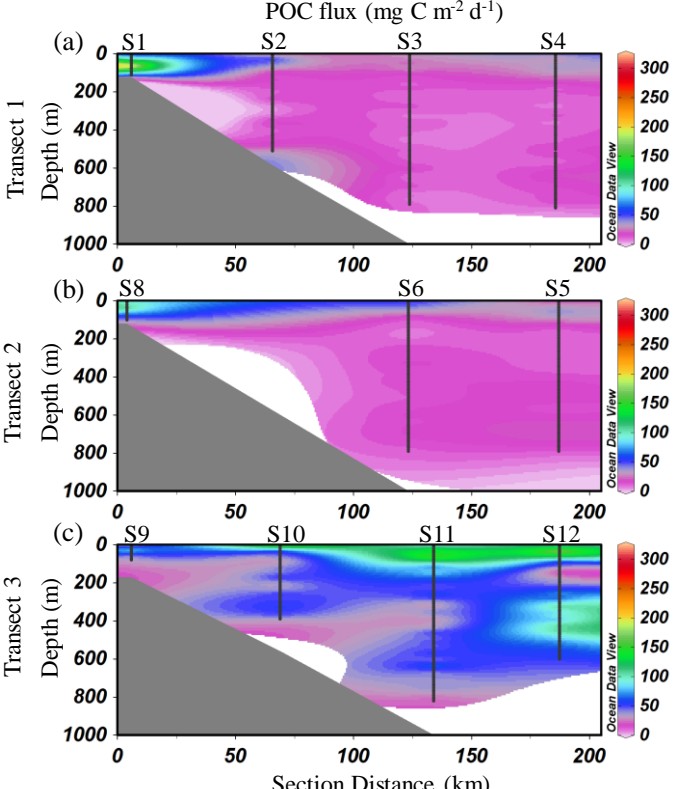

**Fig. 7 Vertical distribution of POC flux across the three transects (mg C m⁻² d⁻¹) in the study area. a: Transect 1; b: Transect 2; c: Transect 3. Interpolated values between S6 and S8 are automatically generated by ODV and should be interpreted with caution.**

The POC flux patterns differed markedly between stations influenced by anticyclonic eddies (S4, S5) and cyclonic eddies (S11, S12) (Fig. 8). In terms of flux magnitude, the highest values at all stations were observed above 100 m. However, POC fluxes at S11 and S12 were consistently higher than those at S4 and S5 across all depths (Fig. 8a). Regarding flux composition, the contribution of small particles to the POC flux was generally higher at the anticyclonic eddy-influenced stations S4 and S5 than at the cyclonic eddy-influenced stations S11 and S12 (Fig. 8b), whereas the contribution of large particles was greater at S11 and S12 compared to S4 and S5 (Fig. 8c).

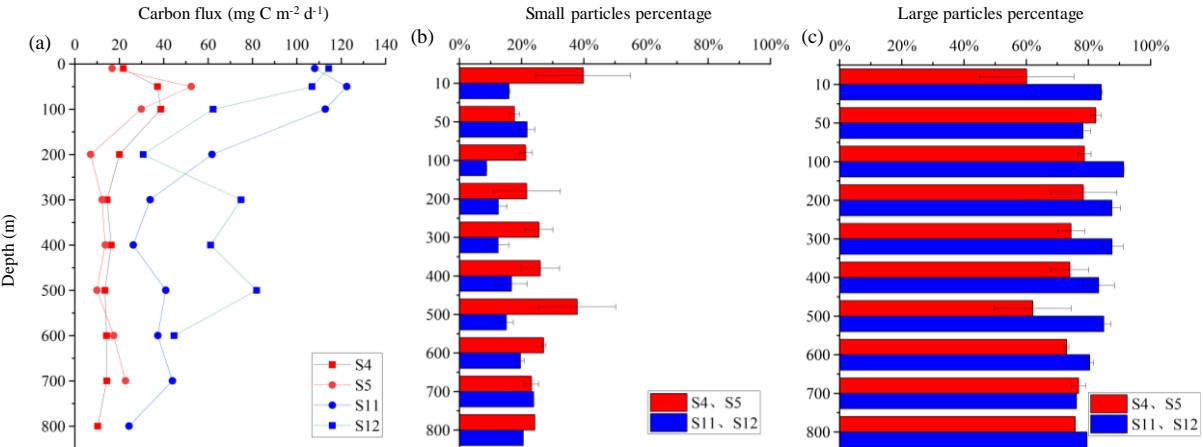

**Fig. 8 POC flux derived from UVP5-HD data in different water layers and contributions of small and large particles in the anticyclonic eddy region (S4, S5) and cyclonic eddy region (S11, S12). a: Carbon flux (mg C m⁻² d⁻¹); b: Small particles percentage; c: Large particles percentage. b, c: mean values and standard deviations of those two stations.**

The results of the Pearson correlation analysis indicate that POC flux does not exhibit significant correlations with most environmental factors in the upper 200 m layer, except for water depth and Chl *a* concentration (Fig. 9a). POC flux shows a significant negative correlation with water depth and a significant positive correlation with Chl *a* concentration. Scatter plot analyses further revealed significant positive relationships between POC flux at 200 m and the water-column integrated Chl *a* concentrations above 200 m (Fig. 9b). For the 400 m and 600 m depth layers, the POC flux showed no significant correlation with the integrated Chl *a* concentration in the upper 200m of the water column (Fig. 9c, d).

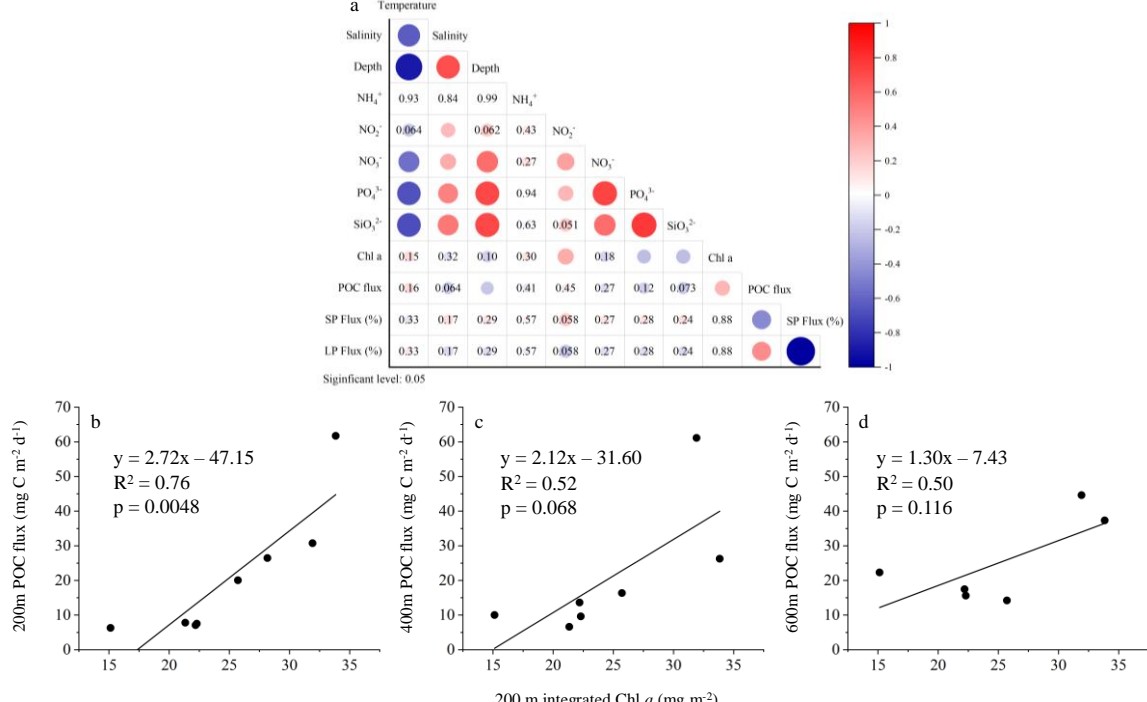

**Fig. 9 Correlation analysis between POC flux and environmental factors.**

**a: Heatmap showing Pearson correlation coefficients (*r*) between POC flux and environmental variables in the upper 200 m. Significant correlations (*p* < 0.05) are highlighted by filled circles with color intensity proportional to *r*, but the correlation coefficient values are not labeled. Non-significant correlations (*p* ≥ 0.05) are indicated by numeric *r* values only. b-d: Correlation between POC (mg C m$^{-2}$ d$^{-1}$) flux at 200 m, 400 m, and 600 m depth and the integrated Chl *a* concentration (mg m$^{-2}$) in the upper 200m. SP: small particle; LP: large particle. A significance level of *p* < 0.05 was used to determine statistical significance.**

## 4 Discussion

### 4.1 Marine particle distribution and controls: cross-system comparisons and regional characteristics

The particle abundance and PVC observed with UVP5-HD along the SCS slope in this study offer important context for understanding regional particle dynamics. When compared to UVP data from other oligotrophic oceanic regions, our findings fall within the broad range of particle concentrations and volume reported for similar low-nutrient systems (Table 2). This comparison also highlights the role of environmental variability in shaping particle distributions in the oligotrophic SCS. The particle abundance in the SCS continental slope (0–100 m: 25–476 particles L$^{-1}$; 0–800 m: 0–783 particles L$^{-1}$) is comparable to values reported for other oligotrophic regions such as the High Nutrient, Low Chlorophyll (HNLC) areas of the Southern Ocean(0–100 m: 0–500 particles L$^{-1}$; Jouandet et al., 2011), and markedly higher than those observed in the mesopelagic zone (200–1000 m) of the

equatorial Pacific (1–4 particles $L^{-1}$; Pretty, 2019). However, it is notably lower than the particle abundance during an iron-fertilized bloom station in the Southern Ocean ($1400 \pm 200$ particles $L^{-1}$; Jouandet et al., 2011), where

artificial nutrient enrichment significantly boosted particle production. This discrepancy underscores the influence of localized biogeochemical conditions on particle abundance. In terms of particle volume concentration (PVC), the SCS slope exhibits values ranging from 0–6.7 $mm^3$ $L^{-1}$ (mean $\pm$ SD: $0.3 \pm 0.4$ $mm^3$ $L^{-1}$), aligning well with the Gulf of Alaska shelf (0.1–1 $mm^3$ $L^{-1}$; Turner et al., 2017) and the HNLC stations in the Southern Ocean (0–50 $mm^3$ $L^{-1}$; Jouandet et al., 2011). Conversely, these values are lower compared to the iron-fertilized bloom stations in the

Southern Ocean ($183 \pm 34$ $mm^3$ $L^{-1}$; Jouandet et al., 2011), reflecting the significant impact of primary production and aggregation processes. The differences between regions suggest the importance of local environmental factors, including nutrient availability and primary production, in shaping marine particle dynamics.

**Table 2 Comparison of particle abundances and volume concentrations in this study with those in other studies. —:**
**no data. Data from the upper 100 m layer are presented separately to facilitate comparison with previous studies**
410 **using similar sampling depths (e.g. Iversen et al., 2010; Jouandet et al., 2011).**

| Location | Depth (m) | Particle abundance range (mean ± SD) ($L^{-1}$) | PVC range (mean ± SD) ($mm^3\ L^{-1}$) | Reference |
|---|---|---|---|---|
| The North Mediterranean | 0–400 | 0–80 | — | Gorsky et al., 2000 |
| The Ligurian Sea | 0–1000 | 0–1108 | — | Stemmann et al., 2008 |
| Off Cape Blanc, NW Africa | 0–80 | 30–100 | — | Iversen et al., 2010 |
| The HNLC stations in the Southern Ocean | 0–100 | 0–500 | 0–50 | Jouandet et al., 2011 |
| Iron-fertilized bloom station in the Southern Ocean | 0–100 | 1400 ± 200 | 183 ± 34 | Jouandet et al., 2011 |
| Southeast of Kerguelen Island (Southern Ocean) | 0–100 | 90 ± 5 | 0.3 ± 0.1 | Jouandet et al., 2014 |
| The Gulf of Alaska shelf | 0–40 | — | 0.1–1 | Turner et al., 2017 |
| The Equatorial Pacific | 200–1000 | 1–4 | — | Pretty, 2019 |
| The North Pacific Gyre | >1000 | 0.1–0.3 | — | Pretty, 2019 |
| The eastern tropical North Pacific | 160–500 | 1–10 | — | Cram et al., 2022 |
| Station ALOHA (22.75ºN, 158.00ºW) | 0–75 | 50–125 | — | James, 2024 |
| SCS slope | 0–100 | 25–476 (95 ± 71) | 0.0–6.7 (0.6 ± 0.8) | This study |
| SCS slope | 0–800 | 0–783 (68 ± 69) | 0.0–6.7 (0.3 ± 0.4) | This study |

411

Horizontally, particle abundance and PVC exhibited clear cross-shelf gradients, with higher values observed at the inner slope stations, and a decreasing trend toward offshore waters (Fig. 4a-c; Fig. 5a-c). This pattern is consistent with observations from several other oceanic regions. Guidi et al. (2008b) studied the distribution of particles (>100 μm) in the South-Eastern Pacific, and found that particle concentrations were highest over the Peru-Chile continental shelf, and decreased progressively toward the open ocean. Forest et al. (2012) found that particle abundance and volume in the surface layer of southeast Beauford Sea exhibited a decline of approximately two orders of magnitude when progressing from the shelf toward the basin. Turner et al. (2017) measured the concentrations of marine particles in the Gulf of Alaska during summer 2015, and found that particle concentrations were highest at stations closest to large inputs of freshwater and glacial discharge, intermediate at some nearshore stations, and lowest over the mid-shelf. Therefore, our results, in line with previous studies, indicate that particle concentrations tend to be higher in nearshore regions. In this study, it was observed that the depth of the nutricline at inner slope stations was shallower than that at outer slope stations (Fig. 3). The shallow nutricline observed at inner slope stations could enhance light availability in the SCM layer, likely promoting higher Chl $a$ concentrations. The distribution of Chl $a$ also clearly indicates that the DCM layer at inner slope stations exhibited higher Chl $a$ concentrations compared to outer slope stations (Fig. 3j-l). The resulting increase in phytoplankton biomass may stimulate organic matter production and particle aggregation (Panaïotis et al., 2024), thereby contributing to elevated particle concentrations there. Additionally, nearshore regions are more susceptible to terrestrial inputs. On one hand, terrestrial input can supply nutrients, stimulating primary productivity and particle formation (Turner et al., 2017). On the other hand, the proximity to the continental margin increases exposure to lithogenic inputs from terrestrial sources (Liu et al., 2016), further enhancing particle loads in inner slope waters. Therefore, from a horizontal perspective, the combined influence of terrestrial inputs and a shallower nutricline in inner shelf regions enhances biogeochemical processes that lead to higher particle concentrations there.

In addition to identifying patterns of particle distribution that are consistent with observations from other oceanic regions, we also observed several localized and unique features specific to this region. The most notable one is the elevated particle abundances and PVC observed at stations S11 and S12 (Fig. 4c, Fig. 5c). At stations S11 and S12, surface particle abundance and PVC reached as high as 130 particles $L^{-1}$ and 0.45 $mm^3$ $L^{-1}$, respectively, substantially higher than those observed at the inner slope station S9, which recorded 80 particles $L^{-1}$ and 0.18 $mm^3$ $L^{-1}$. Given their offshore location (> 200 km from land), the particle enrichment at these sites is unlikely to stem from terrestrial inputs and instead reflects localized biological production. Mesoscale eddies are known to influence upper-ocean nutrient supply, productivity levels, and the physical structure of the water column (Maiti et al., 2008). Cyclonic eddies, in particular, induce upward vertical transport that enhances nutrient supply to the euphotic zone, stimulating phytoplankton growth and aggregate formation (Kahru et al., 2007). Furthermore, convergence and retention zones along the eddy periphery can trap suspended and sinking particles, facilitating localized particle accumulation (Accardo et al., 2025). In our study, the elevated particle concentrations at S11 and S12 support this mechanism, suggesting a coupling between eddy-driven nutrient enhancement and biological particle production. Another localized and distinct observation was the exceptionally high particle abundance detected in the bottom layers at stations S2 and S10 (Fig. 4a, c). At the bottom layer at these two stations, the particle population was overwhelmingly dominated by small particles, which accounted for more than 99.8% of the total abundance (Fig. 4d, f). This explains why the exceptionally high particle numbers did not correspond to elevated PVC values there. Given the low particle concentrations observed in the upper water column at these two

stations, it is unlikely that the elevated particle abundance in the bottom layer resulted from vertical settling from above. The low abundance of large zooplankton observed in this area (Supplementary Fig. S2a, c) does not support the presence of a substantial amount of living particles in the bottom layer. One plausible explanation for this deep particle enrichment is the presence of intermediate nepheloid layers (INLs). INLs are formed by suspended particles spreading along isopycnal surfaces and are commonly found near the edges of continental shelves, slopes, and seamounts (Oliveira et al., 2002). In the SCS, they are primarily formed as a result of resuspension and lateral transport of fine particles along the continental slope (Jia et al., 2019; Chen et al., 2024). The composition of INLs typically consists of a mixture of fine-grained lithogenic particles, resuspended sediments, and biogenic detritus (Oliveira et al., 2002). The exceptionally high particle abundance observed here is more likely derived from sediment resuspension rather than from vertical export originating in the euphotic layer. In contrast, a different case was observed at station S4. A slight elevation in particle abundance was found in the upper 100 m (Fig. 4a), and image analysis confirmed that this layer was also associated with an elevated concentration of large zooplankton (Supplementary Fig. S2a). Given that this station was sampled during nighttime (Table 1), it is plausible that this particle peak resulted from diel vertical migration, during which zooplankton ascend toward the surface at night and contribute to increased particle signal via biological presence and pellet production (Hays, 2003). This observation suggests that in addition to physical processes such as eddy dynamics and lateral resuspension, vertical migratory behavior of zooplankton may also modulate fine-scale particle distributions.

**4.2 Numerical dominance of small particles vs. biogeochemical significance of large particles**

The size composition of marine particles is critically important, as it governs their sinking efficiency, with smaller particles typically sinking much more slowly than larger ones (Clements et al., 2023). In addition, vertical variations in particle size composition can offer insights into particle degradation and transformation processes occurring within the water column (Cram et al., 2022). Using data from the UVP5-HD, this study characterizes the size composition of marine particles along the SCS slope. Small particles contributed over 98% to the total particle abundance in most areas (Fig. 4d-f), highlighting their overwhelming numerical dominance. Slight decreases in the relative contribution of small particles were only observed in certain high-productivity regions, such as the nearshore stations of Transects 1 and 2, and stations S11 and S12 along Transect 3. Due to the lack of image data for small particles, we were unable to determine the exact cause of the decreased proportion of small particles in these areas. The study area represents a typically oligotrophic environment. Previous studies have reported low phytoplankton productivity in this region, with the community largely dominated by picophytoplankton (Zhang et al., 2023). The low biomass and small size of phytoplankton cells limit their ability to form large particles or aggregates through aggregation processes. Additionally, low primary productivity is often accompanied by low zooplankton abundance (Liu et al., 2020). The low biomass of large-sized zooplankton further limits the production of large particles, contributing to the dominance of small particles in abundance.

Although small particles overwhelmingly dominated in terms of abundance, the pattern was notably different when PVC was considered. Across the entire study area, small particles contributed between 13% and 74% of the total PVC (mean ± SD: 39% ± 12%) (Fig. 5d-f), whereas large particles accounted for a substantially higher share, ranging from 26% to 87% (mean ± SD: 61% ± 12%) (Fig. 5g-i). This contrast became even more pronounced when examining carbon fluxes. Small particle-derived POC flux accounted for 5% to 77% of the total, with an average contribution of 24% ± 12%, whereas large particles contributed the majority of carbon export, ranging from 23% to 95%, with a mean of 76% ± 12% (Supplementary Fig. S3). This apparent disconnect between numerical dominance and carbon export efficiency underscores the importance of particle size in determining

vertical carbon flux (Cram et al., 2018). Smaller particles tend to have lower sinking velocities due to their higher surface-area-to-volume ratios and reduced mass, which makes them more susceptible to microbial degradation, grazing, and disaggregation in the upper ocean (Riley et al., 2012; Durkin et al., 2015). In contrast, large particles—such as fast-sinking marine snow aggregates, fecal pellets, and large phytoplankton or zooplankton carcasses—settle more rapidly and are therefore more efficient vehicles for transporting organic carbon to depth (Iversen et al., 2010; Forest et al., 2012). In our study, the dominance of small particles in abundance but their relatively minor role in POC flux suggests that carbon export efficiency in the SCS slope region is largely regulated by the production and fate of large, fast-sinking particles.

The vertical partitioning of POC flux between small particles and large particles also revealed distinct depth-dependent patterns. In the upper 200 m, small particles contributed an average of 19% ± 9% to the total POC flux, while large particles dominated with 81% ± 9%. Below 200 m, the relative contribution of small particles increased to 28% ± 12%, whereas that of large particles declined to 72% ± 12%. The enhanced contribution of small particles at depth was statistically significant (t-test, $p < 0.05$). This trend suggests that, although large particles dominate carbon export in the upper ocean, small particles become increasingly important in the mesopelagic zone. Several mechanisms may account for this shift. Large particles, such as fecal pellets and marine snow aggregates, typically sink rapidly and can reach depth with relatively little degradation (Turner, 2015). However, they are also more prone to fragmentation and microbial decomposition during descent, especially in the upper mesopelagic zone (Stamieszkin et al., 2017). As these large particles disaggregate, they contribute to the pool of smaller, slower-sinking particles, thereby increasing the relative contribution of small particles to the total flux at greater depths. In contrast, small particles—although less efficient in transporting carbon due to their slower sinking velocities and higher residence times—can persist longer in the water column. On the other hand, as discussed in Section 4.1, nepheloid layers were observed in the bottom waters along the SCS slope. This process can introduce substantial amounts of fine sediment particles into the water column through resuspension and lateral transport (Zhou et al., 2020; Chen et al., 2024), further contributing to the increased proportion of small particles in the mesopelagic POC flux. Therefore, the elevated contribution of small particles to POC flux in the mesopelagic layer may reflect not only the progressive disaggregation of larger particles during sinking, but also the influence of resuspended fine sediments associated with nepheloid layers. These observations highlight the dynamic nature of particle flux attenuation and transformation with depth in this region.

**4.3 Eddy-driven variability in particle dynamics and carbon export**

Mesoscale eddies are highly frequent in the SCS, with 230 eddies generated on average each year (Jin et al., 2024). These eddies exert a profound influence on regional biogeochemical processes (Xiu and Chai, 2011; Guo et al., 2015; Xu et al., 2025). Accounting for their impact is essential when examining particle characteristics and carbon export in this dynamic region. In this study, we compared particle characteristics between two contrasting eddy regimes: a cyclonic eddy (stations S11 and S12) and an anticyclonic eddy (stations S4 and S5). Across most of the water column (0–800 m), PVC was markedly higher in the cyclonic eddy region than in the anticyclonic region (Fig. 6a). This pattern likely reflects the enhanced nutrient availability and elevated phytoplankton biomass observed under cyclonic eddy influence, which favored the formation of biogenic particles.

In addition to PVC, the two eddy types also exhibited distinct particle size structures (Fig. 6b-e). In the anticyclonic eddy region, small particles generally contributed more than 40% to the total PVC throughout the water column, with average contributions of 45% ± 6% at station S4 and 55% ± 10% at station S5. In the cyclonic eddy region, the proportion of small particles was noticeably lower, with average contributions of 35% ± 8% at

station S11 and 30% ± 7% at station S12, typically remaining below 40% throughout the water column. This contrast indicates that the two types of mesoscale eddies exert different influences on the particle size composition. Although this study did not include direct analysis of phytoplankton community composition, previous studies have shown that cyclonic eddies could promote the growth and proliferation of large-sized phytoplankton, such as diatoms, allowing them to become dominant within the community (Shih et al., 2020; Chenillat et al., 2024). These larger phytoplankton are more prone to form aggregates through intercellular coagulation (Panaïotis et al., 2024), thereby contributing to the generation of larger particles. Moreover, they provide a favorable food source for large zooplankton (Mangolte et al., 2022), which were indeed observed in greater abundance at stations S11 and S12 in this study (Supplementary Fig. 2c). Together, these mechanisms likely contributed to the higher proportion of large particles in the total PVC in the cyclonic eddy region. In contrast, anticyclonic eddies, characterized by nutrient depletion, tend to favor the proliferation of smaller phytoplankton groups, such as picophytoplankton (Dai et al., 2020; Zhang et al., 2023). Compared to diatoms and other large phytoplankton groups, picophytoplankton are less likely to form aggregates through cell coagulation (Guidi et al., 2009). Instead, they are more readily consumed by small zooplankton such as ciliates, channeling biomass into the microbial loop rather than into fast-sinking particles (Honjo et al., 2008). The abundance of large zooplankton at stations S4 and S5 was also relatively low (Supplementary Fig. S2a, b). These conditions weaken the formation of large particles here, thereby contributing to the increased proportion of small particles. Therefore, cyclonic and anticyclonic eddies in the SCS exert different influences on particle concentration and size composition by altering nutrient availability and phytoplankton production processes. Cyclonic eddies tend to promote the formation of large particles, while anticyclonic eddies are associated with an increased proportion of small particles.

How do changes in particle concentration and size composition between cyclonic and anticyclonic eddy regions affect POC export flux? In this study, we applied the method proposed by Guidi et al. (2008a), which estimates POC flux from UVP-derived particle size spectra using an empirical relationship originally developed from a global sediment trap dataset. This method has since been widely applied across various oceanic regions for consistency and comparability (Ramondenc et al., 2016; Clements et al., 2023; Wang et al., 2004a). We acknowledge that regional differences may influence the absolute values of the estimated fluxes, the method remains robust for evaluating relative differences among stations within this study. In terms of carbon flux, the POC export at the cyclonic eddy influenced stations (S11 and S12) was significantly higher than that at the anticyclonic eddy influenced stations (S4 and S5) (Fig. 8a, $t$-test, $p < 0.01$). This difference was pronounced in the upper water column. For example, at 50 m depth, the POC flux at station S11 reached as high as 122 mg C m$^{-2}$ d$^{-1}$, while values at stations S4 and S5 were only 37 and 52 mg C m$^{-2}$ d$^{-1}$, respectively (Fig. 8a). A similar pattern was observed in the mesopelagic layer below 200 m, where fluxes at S11 and S12 remained markedly higher than those at S4 and S5. Quantitatively, the POC flux in the cyclonic eddy region was more than twice that in the anticyclonic eddy region. Cyclonic eddies can enhance POC export by stimulating primary productivity in the upper water column, whereas anticyclonic eddies tend to suppress it (Li et al., 2017; Zhang et al., 2020; Zhou et al., 2020). The significant correlation observed in this study between POC flux and Chl $a$ (Fig. 9) further supports this relationship, indicating that increased phytoplankton biomass due to mesoscale processes is conducive to higher carbon export. An additional insight of this study is that the contribution of small particles to the POC flux was higher in the anticyclonic eddy influenced region compared to the cyclonic eddy influenced region, whereas the opposite pattern was observed for large particles (Fig. 8b, c). This pattern highlights the distinct export pathways associated with different eddy regimes. In anticyclonic eddies, although small particles still contribute

less to the POC flux than large particles, their relative contribution is noticeably higher compared to that in cyclonic eddies. In contrast, cyclonic eddies enhance the formation of large, fast-sinking particles (Fig. 6), leading to a reduced proportional contribution of small particles to vertical carbon export. These findings emphasize the role of eddy polarity in shaping not only carbon flux magnitude, but also its particle size structure.

## 5 Conclusion

This study represents one of the first efforts to apply high-resolution, size-revolved observations from the UVP5-HD to investigate depth-dependent particle distributions and carbon export dynamics along the continental slope of the SCS. The results reveal distinct spatial and vertical heterogeneity in particle distribution, size structure,
and POC flux, strongly modulated by mesoscale eddy activity. Cyclonic eddies were associated with elevated particle concentrations, enhanced contributions from large particles, and significantly higher POC fluxes throughout the upper and mesopelagic layers. In contrast, anticyclonic eddies exhibited a higher proportion of small particles and reduced carbon export efficiency. Although small particles dominated numerically, large particles contributed disproportionately to the total carbon flux due to their greater volume and faster sinking
velocities. These findings demonstrate the importance of revolving particle size composition when evaluating the biological carbon pump. By coupling UVP-derived observations with eddy-revolved physical features, this study provides novel insights into the mechanisms linking mesoscale processes to particle-mediated export, and enhances our understanding of carbon transport dynamics in oligotrophic marginal seas like the SCS.

### Data Availability Statement
Data available on request from the authors.

### Author Contribution
SG: Conceptualization, Data curation, Formal analysis, Methodology, Software, Visualization, Writing-original draft preparation. MZ: Data curation, Formal analysis, Investigation, Methodology, Validation. WX: Data curation, Methodology. SZ: Investigation. SL: Data curation, Investigation. YW: Data curation, Investigation. JD: Data
curation. CZ: Visualization. XS: Conceptualization, Data curation, Funding acquisition, Project administration, Supervision, Writing-review&editing.

### Competing interests

The authors declare that they have no conflict of interest.

### Disclaimer

Publisher's note: Copernicus Publications remains neutral with regard to jurisdictional claims made in the text, published maps, institutional affiliations, or any other geographical representation in this paper. While Copernicus Publications makes every effort to include appropriate place names, the final responsibility lies with the authors.

### Acknowledgements
We thank the crew and captain of the R/V *Nanfeng* for the logistic support during the cruise. We thank Dr. Chuanjun

Du for his assistance with data analysis. This work was supported by the National Key Research and Development Program of China (No. 2024YFE0114300), the National Natural Science Foundation of China (No. 32371619, U2006206), the National Basic Research Program of China (No. 2014CB441504), the International Partnership Program of Chinese Academy of Sciences (No. 133137KYSB20200002, 121311KYSB20190029).

**Financial support**

This research has been supported by the National Key Research and Development Program of China (No. 2024YFE0114300), the National Natural Science Foundation of China (No. 32371619, U2006206), the National Basic Research Program of China (No. 2014CB441504), the International Partnership Program of Chinese Academy of Sciences (No. 133137KYSB20200002, 121311KYSB20190029).

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
