# Peer review of "Distribution and fluxes of marine particles in the South China Sea continental slope: implications for carbon export"

_EGUsphere, 2025_

## Author Comment (AC1)

**Dear reviewer,**

**Thank you very much for your valuable suggestions. Your comments have been extremely helpful, and we have done our best to revise the manuscript accordingly. We hope that the revised version meets your expectations. The following is a detailed account of the specific revisions.**

1. General comments

Your article and the questions you raised are interesting. However, I have the impression that you could learn more from the UVP data. I don't really understand why living particles aren't separated out and studied separately from non-living particles. I know this is the method used by Kiko et al. 2022, but I would have liked a few sentences explaining why we can make the assumption that living particles can be counted as non-living particles. Moreover, in Kiko et al. 2022 the vast majority of particles are indeed non-living (by image visualization), do you have any idea of the percentage of living particles imaged in your dataset? Depending on the location, this could be very high.

This distinction would allow several things to be done, depending on the level of classification you use:

Living/non-living classification: confirmation of physical processes on biology, not only via chl-a as you are currently doing, but also on the distribution of zooplankton. It was for example done in Trudnowska et al. 2021 or Perhirin et al. 2025.

Living/non-living classification according to shape of the particles: this would make it possible to refine the calculation of biomass or the approximation currently made with the ESD.

Response: Thank you for your thoughtful and constructive comments. We appreciate your suggestion regarding the separation of living and non-living particles and the potential value of such classification. In our study, we used the UVP5.0-HD system, which generates vignette images only for particles with an ESD greater than approximately 500 μm. Therefore, it is not technically possible to classify smaller particles (ESD < 500 μm) as living or non-living, as they are not imaged individually. While classification of larger particles is feasible, our study primarily focuses on the distribution, size composition, and carbon flux of the entire particle spectrum in response to mesoscale eddies. Small particles account for a substantial portion of total particle abundance and volume and cannot be excluded from the analysis. For this reason, and in line with many previous UVP-based studies (e.g., Ramondenc et al., 2016; Fender et al., 2019; Clements et al., 2023), we treated all detected particles together without separating living and non-living fractions.

Regarding your concern that his may affect the accuracy of carbon flux estimates: we used the empirical parameterization developed by Guidi et al. (2008), which relates total particle size spectra (including both living and non-living components) to sediment trap-derived POC fluxes at a global scale. This approach has been widely adopted and validated, and does not require prior separation of particle types. The only potential source of uncertainty in this method lies in whether the empirical parameters derived from the global dataset in Guidi et al (2008) are fully applicable to the SCS. This issue has been addressed in the Discussion section.

That said, we agree that studying the biological component of the particle pool—especially zooplankton—is a valuable direction. Inspired by your suggestion, we analyzed the zooplankton vignettes from large particles identified from UVP data and observed elevated zooplankton abundance in eddy-affected regions. We have now included these results in the revised manuscript and expanded the discussion to address potential eddy effects on zooplankton distribution. However, the primary focus of

this study is on particles. The zooplankton-related findings, while valuable, will be explored more comprehensively and in greater detail in a separate manuscript we plan to prepare.

Finally, we truly appreciate your expertise reflected in the review, and your insightful comments suggest a strong expertise in this field. If possible, we would be very interested in the opportunity to collaborate with you in the future, especially on the zooplankton dynamics with UVP study in the western Pacific. If you are interested, please feel free to contact me or just leave your message. Thanks.

Clements D J, Yang S, Weber T, McDonnell A M P, Kiko R, Stemmann L, Bianchi D. 2023. New estimate of organic carbon export from optical measurements reveals the role of particle size distribution and export horizon. Global Biogeochemical Cycles, 37, e2022GB007633.

Fender C K, Kelly T B, Guidi L, Ohman M D, Smith M C, Stukel M R. 2019. Investigating particle size-flux relationships and the biological pump across a range of plankton ecosystem states from coastal to oligotrophic. Front. Mar. Sci. 6: 603.

Guidi L, Jackson G A, Stemmann L, Miquel J C, Picheral M, Gorsky G. 2008. Relationship between particle size distribution and flux in the mesopelagic zone. Deep-Sea Research I, 55: 1364-1374.

Ramondenc S, Goutx M, Lombard F, Santinelli C, Stemmann L, Gorsky G, Guidi L. 2016. An initial carbon export assessment in the Mediterranean Sea based on drifting sediment traps and the Underwater Vision Profiler data sets. Deep-Sea Research I, 117: 107-119.

Wang Z Y, Fang C, Yang C H, Zhang G Y, Sun D. 2024. Latitudinal gradient and influencing factors of deep-sea particle export along the Kyushu-Palau Ridge in the Philippine Sea. Science of the Total Environment, 906: 167460.

Next, two of your deployments were done by night (S2 and S4), could the presence of DV migrators have affect your results?

Response: Thank you for this important observation. We acknowledge that diel vertical migration of zooplankton may influence the abundance and composition of large particles observed in night-time deployments. In our study, two stations (S2 and S4) were indeed sampled during nighttime. Although our dataset is not designed to specifically evaluate the effect of DVM, we now discuss this potential bias in the revised manuscript and acknowledge it as a possible source of uncertainty in the interpretation of particle abundance and POC flux at these stations.

I also feel that there are a number of missing elements that detract from the manuscript's readability. For example, the methodology used for some the analyses is very opaque, the total number of particles observed is never indicated, etc. Some figures are not very clear. I detail these in the following comments.

Response: Thanks for the comment. In the revised manuscript, we have clarified the analytical procedures, added missing information such as the total number of particles observed (now included in Table 1), and improved the quality of several figures based on your comments. We hope this revision improve the readability of our work.

2.  Specific comments

Abstract

Line 21: precise which deviation you used

Response: Thank you for pointing this out. We used the standard deviation (SD) in the expression "68 ± 69 particles $L^{-1}$". We have now clarified this in the revised abstract.

Line 25: you might want to give the mean and deviation as for the particle abundances.

Response: Thank you for the helpful suggestion. We have now added the mean and standard deviation for the estimated POC fluxes, consistent with the presentation of particle abundances. The revised sentence in the abstract reads: "Estimated POC fluxes ranged from 3.4 to 302.4 mg C $m^{-2}$ $d^{-1}$ (mean ± SD: 33.6 ± 34.9 mg C $m^{-2}$ $d^{-1}$), with maxima in the upper 100 m and at stations influenced by cyclonic eddies.".

Lines 25/26: your data 'only' covers two stations influenced by an anticyclonic eddy and two in the periphery of a cyclonic eddy, I don't think that's enough to be able to generalize or to use the plural form of eddy in these two sentences.

Response: Thank you for this observation. We agree that our data coverage is limited, with only two stations associated with an anticyclonic eddy and two at the periphery of a cyclonic eddy. We have revised the relevant sentences in the abstract to avoid overgeneralization and removed the plural form of "eddy". The revised version now reads: "with maxima in the upper 100 m and at two stations located at the periphery of a cyclonic eddy. In contrast, at two stations influenced by an anticyclonic eddy, particle concentrations and export fluxes were reduced, likely due to enhanced stratification and nutrient limitation.".

Graphical abstract

I don't find your graphical summary very useful, it's not very clear. It also lacks a legend to give some context or explanation. I don't understand why the particles in the two eddies are identical. You've also included bacteria and phytoplankton, which are not visible under UVP (although I understand that phytoplankton is approximated by Chl-a in your study). It's not very easy to see that the abundance of small dots is the main difference between the two eddies. I am not sure about the downwelling eddy arrows. The sun is unnecessary. Maybe put some green shapes for chl-a or phytoplankton.

Response: Thank you for your detailed feedback regarding the graphical abstract. We agree that it did not effectively convey the core findings of the study and may have led to some confusion. Based on your suggestion, we have decided to remove the graphical abstract in the revised submission to maintain clarity and focus.

Introduction

I think you can try to focus a little bit more your introduction on the links between eddies and export, I especially regret some references that you cited only in the Discussion and that are in the topic and in the SCS (e.g. Liu et al. 2020).

Response: Thank you for this valuable suggestion. In the revised manuscript, we have rewritten the Introduction to better highlight the scientific rationale and objectives of this study. A new paragraph has been added to specifically introduce how eddies influence biogeochemical dynamics in the ocean.

Line 45: the beginning of the line is not very clear.

Response: Thank you for pointing this out. We have revised the sentence for clarity and now rephrased it as follows: "Therefore, understanding the distribution, size composition, and associated carbon export flux of marine particles is critical for assessing the efficiency of the biological carbon pump.".

Line 50: find a more appropriate reference (one using more sediment traps or at a more global scale).

Response: Thank you for the helpful suggestion. In response, we have replaced the previous reference with two more appropriate studies: Honjo et al. (2008), which provides a global synthesis of sediment trap data, and Harms et al. (2021), which focuses on particle flux variability using sediment traps in the South Indian Ocean.

Lines 58/60: add references (maybe Trudnowska et al. 2021).

Response: Thank you for the suggestion. We have now added the recommended reference (Trudnowska et al., 2021) as well as Trudnowska et al. 2023 to support the statements.

Line 94: maybe try to convert it in mg C m$^{-2}$ d$^{-1}$.

Response: Thank you for the suggestion. We have converted the values reported by Hong et al. (2021) from mmol m$^{-2}$ d$^{-1}$ to mg C m$^{-2}$ d$^{-1}$ for consistency and clarity. The revised sentence now reads: "Hong et al. (2021) quantified cross-shelf POC export in the northern SCS shelf using $^{234}$Th-based estimates and sediment traps, reporting a vertical POC flux of 312 mg C m$^{-2}$ d$^{-1}$ from the euphotic zone and a cross-shelf transport flux of 119 mg C m$^{-2}$ d$^{-1}$.".

Material and methods

Line 119: three transects that are called 《sections》 in Fig. 1.

Response: Thank you for the helpful comment. Accordingly, we have revised Fig. 1 to replace "Section 1/2/3" with "Transect 1/2/3" to ensure consistency with the terminology used in the main text.

Lines 123/124: do you have any reference to cite on the multivariate optimal interpolation algorithm? Or can you at least give some details, because currently it is not very transparent nor reproducible.

Response: Thank you for your comment. The multivariate optimal interpolation algorithm used for generating the Level-4 Sea Surface Salinity (SSS) product is based on the method developed and described in detail by Buongiorno Nardelli et al. (2016), and more recently implemented regionally in Sammartino et al. 2022. We have cited the following references in the revised manuscript.

Buongiorno Nardelli, B., Droghei, R., Santoleri, R.: Multi-dimensional interpolation of SMOS sea surface salinity with surface temperature and in situ salinity data. Remote Sens. Environ., 180, 392–402, http://dx.doi.org/10.1016/j.rse.2015.12.052, 2016.

Sammartino, M., Aronica, S., Santoleri, R., Nardelli, B. B.: Retrieving Mediterranean Sea surface salinity distribution and interannual trends from multi-sensor satellite and in situ data. Rmote Sens., 14, 2502, https://doi.org/10.3390/rs14102502, 2022.

Fig.1: maybe colours the cyclonic/anticyclonic stations or indicate their position with a different shape.

Response: Thanks for your comment. To clarify the eddy-related station categorization, we have revised the figure and caption to explicitly state that red dots indicate stations influenced by the anticyclonic eddy, while blue dots represent stations located at the periphery of the cyclonic eddy.

Table 1: add the units for longitude, latitude, time (time zone). Please clarify if UVP deployment depth is the maximum depth of deployment or not. Add the total number of particles and the volume sampled per deployment. Why there is no station 7? You never mentioned or explained it.

Response: Thank you for the detailed and helpful comments on Table 1. In response:

We have added the appropriate units for latitude (°N), longitude (°E), and time (UTC + 8). We clarified in the table caption that "UVP deployment depth" refers to the maximum depth reached during each deployment. The total number of particles detected and the sampled volume per deployment have been added to the table. Station 7 was omitted due to instrument malfunction.

Line 139: add units for temperature, salinity and pressure.

Response: Thank you for the suggestion. We have now added the units for temperature (°C), salinity (psu), and pressure (dbar) in the description of CTD measurements for clarity.

Line 154: which version of the UVP5 you used? HD? SD?
Response: Thank you for pointing this out. We used the UVP5-HD version in this study. This information has now been added to the revised manuscript.

Line 156: the volume sampled by the UVP5 is 1.053 l per frame, please correct it.
Response: Thank you for pointing out this error. We have corrected the per-frame sampling volume of the UVP5 to 1.053 L in the revised manuscript.

Line 160: which custom software did you used? Can you give some details about it?
Response: We thank the reviewer for the constructive suggestion. In this study, the particle images collected by the UVP 5 were processed using the standard software suite developed by the Laboratoire d'Océanologie de Villefranche-sur-Mer, including ZooProcess (v7.22) and PkID (v1.26). These tools are widely used in UVP data processing workflows.

Line 164: do you have any idea on how the ESD approximation could have impact your volume concentration result? If you upload your images in ecotaxa, major axis and minor axis are computed for each particle and thus it is possible to use an ellipsoidal shape (that would maybe be more accurate than the ESD), or maybe your personal software can do it.
Response: Thank you for raising this important point. In our study, particle volume was estimated based on the equivalent spherical diameter (ESD), assuming spherical geometry, which is a commonly used approach in UVP-based analyses (e.g., Guidi et al., 2008; Fender et al., 2019; Wang et al., 2024). We acknowledge that his assumption introduces uncertainty, especially for irregularly shaped particles such as marine snow aggregates and fecal pellets. While we agree that using major and minor axes to derive ellipsoidal volumes (e.g., via Ecotaxa) can improve accuracy for individual particles, our current instrument (UVP5-HD) only records axis data for particles with extracted vignette images—typically the larger particles ($\geq \sim 500\ \mu m$). For the majority of smaller particles, no image or shape data are available, thus precluding the consistent application of ellipsoidal geometry across the full size spectrum. Given this constraint, we retained the widely used ESD-based spherical approximation for continuity and comparability with previous studies.

Fender C K, Kelly T B, Guidi L, Ohman M D, Smith M C, Stukel M R. Investigating particle size-flux relationships and the biological pump across a range of plankton ecosystem states from coastal to oligotrophic. Frontiers in Marine Science, 2019, 6: 603.
Guidi L, Gorsky G, Claustre H, Miquel J C, Picheral M, Stemmann L. Distribution and fluxes of aggregates > 100 μm in the upper kilometer of the South-Eastern Pacific. Biogeosciences, 2008, 5: 1361-1372.
Wang X Y, Li H L, Zhang J J, Chen J F, Xie X H, Xie W, Yin K D, Zhang D S, Ruiz-Pino D, Kao S J. Seamounts generate efficient active transport loops to nourish the twilight ecosystem. Science advances, 2024, 10: eadk6833.

Lines 168/170: in ESD?
Response: Thank you for pointing this out. We confirm that both the upper (1.5 mm) and lower (100 μm) size limits used for particle flux calculations refer to the ESD. This has now been clarified in the revised manuscript.

Line 172: as explained before, the absence of classification between living and non-living particles is the main problem for me. Can you detail a little bit this choice? It would also be nice to have an idea of the percentage of living particles among the total amount of particles.

Response: Thank you for this thoughtful comment. As mentioned earlier, the UVP5.0 system used in this study only capture vignette images for particles with an ESD greater than approximately 500 μm. For smaller particles, only size and abundance are recorded, and no image data are available. As a result, the classification of particles into living and non-living categories is only possible for the larger particle fraction.

The main focus of our study is to investigate the abundance, size structure, and carbon flux of the overall particle field in relation to physical dynamics. While zooplankton dynamics are indeed interesting, they were not the primary research target at the outset. We initially intended to address the classification and distribution patterns of zooplankton in a separate manuscript. However, we appreciate your suggestion, and we have now included zooplankton data extracted from UVP vignette classification in the revised manuscript. Regarding your request for the proportion of living particles, we performed a calculation based on the imaged particles (>500 μm), and found that living particles (primarily zooplankton) accounted for approximately 31% ± 22% (mean ± SD) of the total particle count in this size range on average. Due to space limitations and the scope of the current study, we have chosen not to elaborate extensively on this aspect, but we fully agree it offers a valuable direction for future research.

Line 173: can you please give examples of small and large particles?

Response: Thank you for the suggestion. We acknowledge that particles smaller than 0.5 mm are generally below the UVP5 image resolution threshold and cannot be reliably identified from photographs. As a result, we cannot provide specific examples based on morphology. However, based on prior studies, these small particles likely include fine detritus, small fecal pellets, or other unresolved aggregates. Large particles included marine snow aggregates, sizable fecal pellets, zooplankton carcasses, and other biologically derived material. We followed the size-based classification scheme used by Kiko et al. (2017, 2022), in which particles with ESD < 530 μm are categorized as micrometric (small) and those with ESD ≥ 530 μm as macroscopic (large). We have added a clarification in the revised text.

Kiko R, Biastoch A, Brandt P, Cravatte S, Hauss H, Hummels R, Kriest I, Marin F, McDonnell A M P, Oschlies A, Picheral M, Schwarzkopf F U, Thurnherr A M, Stemmann L. Biological and physical influences on marine snowfall at the equator. Nat. Geosci., 2017, 10: 852-858.

Kiko R, Picheral M, Antoine D, et al. A global marine particle size distribution dataset obtained with the Underwater Vision Profiler 5. Earth Syst. Sci. Data, 2022, 14: 4315-4337.

Lines 174/175: please clarify or justify.

Response: Thank you for pointing this out. We have clarified the rationale behind this sentence. The size-based classification allow us to explore differences in vertical distribution and particle behavior, as larger particles are generally associated with aggregation and faster sinking, whereas smaller particles tend to remain suspended or result from disaggregation. We have revised the sentence accordingly.

Line 187: the formula is not readable, please modify.

Response: Thank you for your comment. To improve readability and avoid rendering issues across platforms, we have reformatted the equation as an image and inserted it into the manuscript.

Lines 188/189: the formula is not readable so my question is a little bit naive, but did you check that the m(d) and w(d) values you get were in the range of observed values for such particles? Or at least in the range of possible values in SCS?

Response: Thank you for pointing this out. We clarify that m(d) denotes the carbon mass per particle (in mg C), and that all variables in the flux equation are converted into consistent units during calculation, resulting in a final flux unit of mg C m$^{-2}$ d$^{-1}$.

Line 197: are A and B coefficients adapted to the SCS? Do you have an idea on how it could impact your results?

Response: Thank you for this important question. The empirical coefficients A and B used in our flux calculations (A = 12.5, B =3.81) were derived from global ocean data by fitting UVP-derived particle size distributions to sediment trap-based carbon fluxes (Guidi et al., 2008). These coefficients have been widely applied in various regions, especially in other oligotrophic systems, such as the tropical Western Pacific and Mediterranean Sea (Ramondenc et al., 2016; Wang et al., 2004a, b). To maintain consistency with previous studies, we adopted the same set of empirical parameters. We acknowledge that regional variability in particle composition, density, and sinking behavior, may cause deviations from the global relationship. A discussion on this limitation has been included in the manuscript, and we will consider developing regionally tuned coefficients in future work as more direct flux measurements become available in the SCS.

Guidi L, Jackson G A, Stemmann L, et al. Relationship between particle size distribution and flux in the mesopelagic zone. Deep-Sea Research I, 2008, 55: 1364-1374.

Ramondenc S, Goutx M, Lombard F, Santinelli C, Stemmann L, Gorsky G, Guidi L. An initial carbon export assessment in the Mediterranean Sea based on drifting sediment traps and the Underwater Vision Profiler data sets. Deep-Sea Research I, 2016, 117: 107-119.

Wang X Y, Li H L, Zhang J J, et al. Seamounts generate efficient active transport loops to nourish the twilight ecosystem. Science Advances, 2024a, 10: eadk6833.

Wang Z Y, Fang C, Yang C H, Zhang G Y, Sun D. Latitudinal gradient and influencing factors of deep-sea particle export along the Kyushu-Palau Ridge in the Philippine Sea. Science of the Total Environment, 2024b, 906: 167460.

Lines 203/205: please give the unit for abundance, volume concentration and POC flux. Were depth profiles generated per station? Aggregated per transect/section?

Response: Thank you for your comment. We have now specified the units for particle abundance (particles L$^{-1}$), particle volume concentration (mm$^3$ L$^{-1}$), and POC flux (mg C m$^{-2}$ d$^{-1}$). We have clarified that depth profiles were generated for each station, and additionally, section plots were constructed to show the along-transect distribution of particle abundance, volume concentration and POC fluxes.

Line 208: add references for Shapiro and Levene tests.

Response: Thank you for the suggestion. We have now added references for the tests in the Methods section.

Lines 209/210: I would avoid using PCA for Pearson correlation analysis, it is usually used for Principal Component Analysis and it can be confusing, especially if people do not read this specific sentence. Either change your acronym, or write 'pearson correlation analysis' each time.

Response: Thank you for pointing this out. We have removed the "PCA" and now refer to "Pearson correlation analysis" in full throughout the manuscript.

Line 222: a certain degree is quite vague.

Response: Thanks for the comment. We have revised the sentence to "Based on the distribution of sea surface salinity (SSS) (Fig. 1c, Supplementary Fig. S1f), the salinity at stations S9 and S10 along Transect 3 was lower than at other stations, indicating a certain degree of influence from the Pearl River plume.".

Fig. 2 and following: maybe try to indicate clearly in the figures where the cyclonic and anti-cyclonic stations were located.

Response: Thanks for the comment. In the revised figure2, we have now indicated the stations influenced by anticyclonic and cyclonic eddies using red and blue colors, respectively. This distinction has also been clearly explained in the updated figure captions.

Lines 232/244: maybe spend less time on nutrients that were not very used in the discussion.

Response: Thank you for the suggestion. We have revised this section to streamline the description of nutrient distributions.

Fig. 4: x-axis unit and name are missing.

Response: Thanks for the comment. We have added the unit and name for x-axis.

Line 266: again precise which deviation you showed.

Response: Thank you for the comment. We confirm that the value reported is the standard deviation (SD), and we have revised the sentence to clearly indicate this.

Line 276: is 'significantly' related to a t-test? If so, precise it along with the p-value?

Response: Thank you for pointing this out. Since no formal statistical test was conducted for this comparison, we have revised the sentence to avoid implying statistical significance.

Line 282: again precise which deviation you showed.

Response: Thanks for the comment. We have revised the sentence to clearly indicate "mean ± SD".

Lines 294/295: how did you choose the images, how they are representative? Moreover, why did you not show any images from the anticyclonic influenced stations? Were particles too small to be imaged? If so, precise it in material and methods.

Response: Thank you for the comment. The images included in the initial version were randomly selected and not intended to support any quantitative analysis. To avoid confusion, we have decided to remove them from the revised manuscript.

Line 295: again precise which deviation you showed.

Response: Thanks for the comment. We have revised the sentence to clearly indicate "mean ± SD".

Fig. 5 and Fig. 6: precise in the caption that the colourbars are different for each section.

Response: Thank you for the comment. Due to the large difference in value ranges among the particle variable (e.g., small particle proportions consistently >95%, large particle proportions <5%), unified colorbars would mask spatial patterns. We have now clarified in the figure caption that colorbars differ between panels and explained the reason.

Fig.7: among the 12 images you show, 4 are recognizable living particles. If this is representative of the images at S11 and S12, then it would mean that one third of the particles you used to compute POC flux are not directly sinking and thus not directly contributing to carbon export.

Response: Thanks for the comment. As we mentioned above, the images were randomly selected and not support any quantitative analysis. We have deleted it. For your worried about the POC flux estimation, we agree that the behavior and contribution of zooplankton to vertical carbon export can be complex and uncertain. As you pointed out, some living particles such as zooplankton may not be directly sinking, or their sinking behavior may be variable due to diel vertical migration or active swimming. However, as noted earlier, our POC flux estimates are based on the empirical approach proposed by Guidi et al. (2008), which integrates all detectable particles (living and non-living) in the size spectrum and was calibrated against sediment trap data across oceanic environments. This method inherently accounts for the mixed nature of the particle pool and the statistical relationship between total particle size distributions and observed fluxes. We acknowledge this as a potential source of uncertainty and have addressed it in the Discussion section of the revised manuscript.

Line 319: are you talking of the mean among stations or per 5-m depth bin? Please clarify.

Line 326: same. Are you talking of the mean among stations or per 5-m depth bin? Please clarify.

Response: Thank you for your comment. We confirm that the values were calculated by aggregating all 5-m binned data across stations, and we have clarified this in the revised sentence.

Line 327: again precise which deviation you are using.

Response: Thanks for the comment. We have revised the sentence to clearly indicate "mean ± SD".

Lines 328/329: the end of the sentence should be part of the discussion and it probably needs some references.

Response: Thanks for the comment. We have removed the interpretive part of the sentence from the Results section and will address these factors more appropriately in the Discussion.

Lines 332/333: same

Lines 339/340: same

Response: Thanks for the comment. We have deleted the contents not belonging here.

Fig. 8: why is there an empty space for what should be station S7 when it was not the case in the previous figures?

Response: Thank you for pointing this out. Station S7 was not included in this study due to a UVP instrument malfunction, and therefore no data are available at that location. We did attempt to modify the earlier UVP-related section plots to match the visual style of Fig. 8, with blank gaps at the missing station. However, the resulting figures were less clear and visually unappealing. For consistency and clarity, we ultimately chose to interpolate across the missing point using ODV settings, resulting in continuous transects.

Lines 349/350: the end of the sentence belongs to the discussion.

Response: Thanks for the comment. We have deleted the contents not belonging here.

Fig. 9: I think that this plot does not show the potential of your results. Could you find a better way to represent these data? Maybe a vertical profile, with 4 coloured paths (one per station)? What is the meaning of high/low? You do not precise it in the text or in the caption. Is there any significant difference between high and low?

Response: Thank you for your helpful comment. Following your suggestion, we revised Fig. 9 to display vertical profiles of POC flux at the four eddy-influenced stations. Each station is now represented by a separate colored line to clearly illustrate differences in the vertical attenuation of flux.

Lines 359/360: this sentence belongs to the discussion.

Response: Thanks for the comment. We have deleted the contents not belonging here.

Line 364: did you realise a statistical test? Does notably mean significantly?

Response: Thank you for your comment. Since no formal statistical tests were conducted for these comparisons, we have revised the text to avoid using the term "significantly" and instead describe the observed differences more cautiously.

Fig. 11: this figure is not clear, is it a bar plot? If so, the caption is wrong, if not, where are the box plots? If too many small values that skew the boxplots close to 0, maybe choose another type of plot that would be more suitable. The colours are unnecessary. What are the dots? (one dot = one station?)

Response: Thank you for your constructive feedback. We agree that the original figure was not sufficiently clear in distinguishing data representation. We have replaced the previous boxplot style figure with a more intuitive format—horizontal bar plots with error bars—representing the mean ± standard deviation of POC flux at each depth across all stations. We believe this updated figure more clearly conveys the vertical distribution patterns of POC flux for total, small, and large particles.

Line 387: again this name is confusing, usually PCA stands for Principal Component Analysis and not Pearson Correlation Analysis, especially if you write again correlation analysis afterwards.

Response: Thanks for the comment. We have changed PCA to Pearson correlation analysis here.

Lines 392/395: sentences belong to discussion.

Response: Thanks for the comment. We have deleted the contents not belonging here.

Lines 397/398: the end of the sentence belongs to the discussion.

Response: Thanks for the comment. We have deleted the contents not belonging here.

Line 400: discussion again.

Response: Thanks for the comment. We have deleted it.

Fig. 12: a) You might want to remove the diagonals. I don't understand why some dots don't have the correlation coefficient or the p-value (not precised) and some have it? I found it strange considering that the largest ones don't have it. Maybe correct the figure or explain it better in the caption. B-d) significant relationships?

Response: Thank you for your comment. We have removed the diagonals in a). In Fig. 12a, only non-significant correlations ($p \geq 0.05$) are labeled with Pearson $r$ values. Significant correlations ($p < 0.05$) are visualized using colored circles without numerical labels, to emphasize their statistical relevance visually. This has now been clarified in the revised figure caption. For b-d, the p-value has been added. The regression at 200 m depth was statistically significant ($p = 0.0048$), while those at 400 m and 600 m were not ($p = 0.068$ and $0.116$, respectively).

Fig. 13: the 0 of a) and b) are different, either you shift it a little bit everywhere or not at all but it is confusing that there is an offset in x-axis and y-axis for a)but only in y-axis for b).

Response: Thank you for pointing this out. As part of the broader revision of the manuscript, Fig. 13 has been removed in the updated version, as it is no longer necessary following the restructuring of the analysis and results.

Line 413: no data was available in the SCS? Even from other methods?

Response: Thank you for this question. To the best of our knowledge, there are currently no published datasets reporting in situ particle abundance or volume concentration in the water column of the South China Sea, especially based on imaging methods such as the Underwater Vision Profiler. Previous studies in this region (e.g., Liu et al., 2014; Zhang et al., 2018; Zhou et al., 2020) have primarily focused on particle fluxes using sediment traps or [234]Th-based approaches, which do not provide quantitative

information on particle abundance or size-resolved volume concentration in the water column. Therefore, we believe that our results represent one of the first efforts to characterize the distribution of suspended particles in the SCS using high-resolution optical methods.

Liu J G, Clift P D, Yan W, Chen Z, Chen H, Xiang R, Wang D X. Modern transport and deposition of settling particles in the northern South China Sea: Sediment trap evidence adjacent to Xisha Trough. Deep-Sea Research I, 2014, 93: 145-155.

Zhang J J, Li H J, Xuan J L, Wu Z Z, Yang Z, Wiesner M G, Chen J F. Enhancement of mesopelagic sinking particle fluxes due to upwelling, aerosol deposition, and monsoonal influences in the Northwestern South China Sea. Journal of Geophysical Research, 124(1): 99-112.

Zhou K B, Dai M H, Maiti K C, Chen W F, Chen J H, Hong Q Q, Ma Y F, Xiu P, Wang L, Xie Y Y. Impact of physical and biogeochemical forcing on particle export in the South China Sea. Progress in Oceanography, 2020, 187: 102403.

Line 417: as in the introduction, maybe try to find a more suitable reference.

Response: Thanks for the comment. We have changed it to a more suitable one "Buesseler et al., 2007".

Line 421: you indicated in the introduction and the results that the SCS is an oligotrophic sea, so why did you use mesotrophic results here? Also, I would have liked a reminder of the oligotrophic status of the SCS here.

Response: Thank you for pointing this out. We agree with your comment and have revised the sentence as following: When compared to UVP data from other oligotrophic oceanic regions, our findings fall within the broad range of particle concentrations and volume reported for similar low-nutrient systems (Table 2). This comparison also highlights the role of environmental variability in shaping particle distributions in the oligotrophic SCS.

Lines 424/427: the data you are comparing to were obtained in the first 100 m (see my comment on the Table 2 below). Also, is that a oligotrophic region?

Response: Thanks for the comment. We have added a line for 0-100 m layer data in Table 2, and we have revised the sentence to clarify the depth range of the SCS particle abundance data and ensure a consistent comparison with previous studies. The HNLC regions are typically considered oligotrophic due to strong iron limitation and low primary productivity. We have clarified it in the sentence. The revised sentence now reads:

The particle abundance in the SCS continental slope (0–100 m: 25–476 particles $L^{-1}$; 0–800 m: 0–783 particles $L^{-1}$) is comparable to values reported for oligotrophic regions such as the High Nutrient, Low Chlorophyll (HNLC) areas of the Southern Ocean(0–100 m: 0–500 particles $L^{-1}$; Jouandet et al., 2011), and markedly higher than those observed in the mesopelagic zone (200–1000 m) of the equatorial Pacific (1–4 particles $L^{-1}$; Pretty, 2019).

Lines 434/435: do you think it could be possible to get chl-a values from other papers to do the same figure as in Fig. 12 b-d) but at a larger scale? I think it would be easier to compare your results to other studies this way.

Response: Thank you for the insightful suggestion. We agree that compiling Chl *a* values from previous studies and reproducing a figure similar to Fig. 12 on a broader scale could help place our results in a wider context. However, after careful review of the relevant literature, we found that most published reports provide none Chl *a* concentrations or only provides over broad and inconsistent depth ranges. These inconsistencies in reporting formats and sampling depths make direct quantitative comparison

difficult. Nevertheless, we appreciate the value of this approach and agree it would be an excellent avenue for future synthesis work as more standardized and high-resolution datasets become available.

Line 436: highlight or maybe just suggest?

Response: Thanks for the comment. We have changed it to 'suggest'.

Table 2: maybe you could add one more line focusing on the 0-100 m to be comparable to Jouandet et al 2011 and Iversen et al 2010.

Response: Thanks for the suggestion. We have added one more line focusing on the 0-100 m in Table 2.

Lines 463/465: can you really affirm that just with Fig. 7, especially without showing images from other stations? I think that more analyses on classified images are necessary to justify this affirmation.

Response: Thank you for your comment. We agree with your assessment. The images shown in Fig. 7 were randomly selected and do not provide a statistically representative basis for that statement. To avoid overinterpretation, we have removed the sentence from the revised manuscript.

Lines 483/485: I agree, but a good way to check this hypothesis would be to plot the vertical distribution of LP, SP and living particles if you decide to classify your images. Or maybe try to find the distribution of zooplankton at the time of the sampling.

Response: Thank you for the helpful suggestion. Following your advice, we incorporated zooplankton data into the revised manuscript and presented their vertical distribution to better support our interpretation.

Line 486: might instead of reflect?

Response: Thanks for your suggestion. We have added 'might" before 'reflect'.

Line 495: I think this paragraph could have come earlier, even if the methods are not the same.

Response: Thank you for your suggestion. We have reorganized and revised the discussion section accordingly.

Lines 503/506: I am not really convinced by the comparisons you are doing with various ocean locations with dynamics that potentially very different from the ones you have in the SCS.

Response: Thanks for the comment. We have deleted this sentence to avoid confusion.

Lines 517/518: justify with a reference please.

Response: Thanks for the comment. We have added the reference 'Cram et al., 2018' here.

Line 524: Steinberg et al 2023 is only about salp faecal pellets, not zooplankton.

Response: Thanks for pointing out this. We have revised it to "Steinberg et al. (2023) found that large zooplankton fecal pellets were……".

Paragraph starting at line 530: I like this paragraph

Response: Thanks.

Lines 546 and followings: I think it would be interesting to link your results to front-enhanced zooplankton dynamics, such as Mangolte et al. 2022 (https://doi.org/10.1093/plankt/fbac010) for example. A lot of papers were published on this subject.

Response: Thank you for the insightful suggestion. Following your recommendation, we have integrated zooplankton data into the revised manuscript and included a preliminary analysis of their distribution in relation to eddy structures. We quite agree that linking zooplankton dynamics to physical features is a

very interesting direction. We plan to conduct a more in-depth analysis using our existing dataset in a future study. We also appreciate the reference to Mangolte et al. (2022), which we have reviewed and cited in the revised manuscript.

Line 569: are zooplankton included in the 'production of organic particles'?
Response: Thanks for the comment. Yes. Our zooplankton data show clearly elevated abundances in the cyclonic eddy region, supporting their role in enhancing the particle field. This has now been clarified in the revised manuscript.

Lines 588 and followings: I like the fact that you included a Data uncertainties paragraph, but the uncertainties you cited might be a little bit too obvious to have a specific paragraph, maybe elaborate.
Response: Thanks for the comment. We have elaborated it in the revised manuscript.

Line 604: "biological production" might be too large considering that zooplankton is not at all included in your study, I would replace it by chlorophyll-a concentration, or anything else focusing only on phytoplankton.
     Response: Thank you for your comment. In the revised version of the manuscript, we have incorporated zooplankton data derived from UVP vignette classification, and observed elevated zooplankton abundance in eddy-influenced regions. Given that both phytoplankton (as inferred from chlorophyll-a) and zooplankton are now considered in our analysis, we believe that the term "biological production" is more justified in this context.

   Once again, thank you very much for your valuable comments. We sincerely hope that our responses and revisions meet your expectations.
   Best regards,
Shujin Guo

---

## Author Comment (AC2)

**Dear reviewer,**

    **Thank you for your insightful suggestions. We have substantially revised and reorganized the manuscript to improve the clarity and coherence of the scientific questions. We sincerely hope that the revised version meets your expectations.**

This study presents data obtained from UVP deployments on a continental slope in a marginal sea, focusing on particle distributions, abundance, POC flux, and potential influencing factors such as mesoscale eddies. The particle imaging data are valuable and have the potential to contribute to the broader biogeochemical and oceanographic community. However, the overall novelty of the study is not readily apparent. The manuscript largely reads as a descriptive study, and many of the conclusions reiterate findings that have already been documented in prior work, including those cited within the manuscript.

**Response:** Thank you very much for your valuable comment. We appreciate your candid assessment. We agree with your opinion that the original presentation may have appeared overly descriptive. In response, we have substantially revised the manuscript, from the Introduction to the Results and Discussion sections, to better highlight the core scientific questions and improve clarity and structure.

    Marine particles or aggregates play a crucial role in oceanic carbon export and the functioning of the biological pump, serving as highly efficient carries of organic carbon to the deep ocean. However, due to inherent difficulties in sampling, their distribution pattern, size characteristics, and controlling mechanisms remain poorly understood. In recent decades, the development of UVP has enabled significant progress in studying particle dynamics in various oceanic regions. Nevertheless, there is still a lack of such observations in the western Pacific, particularly in the SCS. Therefore, characterizing the distribution and properties of marine particles in the SCS and identifying their key drivers represents the first objective of this study. Secondly, mesoscale eddies are a prominent and frequently occurring physical feature in the SCS, and their influence on particle distribution and carbon export is both significant and inevitable. While several previous studies have investigated the impact of eddies on POC export in the SCS using sediment traps, the sediment trap method is limited in the ability to capture high-resolution vertical flux data and provide no information on the size composition of particles. This limitation hampers our ability to fully understand the mechanisms by which mesoscale eddies regulate particle-mediated carbon export. In this context, our use of UVP data offers a new perspective by resolving the vertical structure and size-dependent characteristics of particle fluxes, thereby deepening our understanding of eddy-driven biogeochemical processes in the region.

    Finally, we have revised and reorganized the manuscript according to your valuable comments, and we hope that the updated version meets your expectations.

One notable example is the attribution of the increasing relative contribution of small particles to total POC flux with depth to large particle disaggregation. This interpretation appears speculative and is not sufficiently supported by the presented data. In regions such as the South China Sea, sediment resuspension and lateral transport of particulate matter are also known to influence their contribution to the POC flux. Without additional constraints, it is difficult to disentangle the relative contributions of these processes. In this context, Figure S5 does not clearly demonstrate a vertical trend. I recommend the authors perform statistical analyses (e.g., regression or correlation tests) to better support their interpretations.

**Response:** Thank you for this insightful comment. We agree that the original Fig. S5 was insufficient to support the interpretation regarding the increasing contribution of small particles with depth. In the revised manuscript, we have removed Fig. S5 and restructured the analysis by categorizing the water column into two layers: the epipelagic layer (0-200 m) and the mesopelagic layer (>200 m), instead of using previous depth intervals. Our updated results show that in the upper 200 m, small particles contributed an average of 19% ± 9% to the total POC flux, while large particles accounted for 81% ± 9%. Below 200 m, the contribution of small particles increased to 28% ± 12%, with large particles contributing 72% ± 12%. This increase in the proportion of small particles at depth was statistically significant (t-test, $p < 0.05$). Based on this revised analysis, we have updated the discussion accordingly. Moreover, we appreciate your suggestion regarding other possible mechanisms, such as sediment resuspension and lateral transport, which are known to influence POC flux in the SCS. In response, we have incorporated a discussion of intermediate nepheloid layers and their potential contribution to small particles at depth, to present a more comprehensive interpretation of the observed patterns.

The manuscript would also benefit from improved organization. The rationale and scientific questions driving the study are not clearly articulated in the Introduction, making it difficult to follow the study's objectives and scope. Furthermore, the final portion of the Discussion section, which touches on data uncertainty, lacks clear linkage to the preceding content. A more structured and cohesive progression of ideas is needed to strengthen the overall narrative and scientific discussion.

**Response:** Thank you for your valuable comment. We have revised the Introduction to more clearly articulate the rationale and scientific questions guiding this study, with an emphasis on the ecological significance of marine particles and the role of mesoscale eddies in modulating particle-mediated biogeochemical processes in the SCS. To improve the coherence and logical flow of the Results and Discussion section, we have restructured it around three clearly defined focal points: (1) Marine particle distribution and controls: cross-system comparisons and regional characteristics; (2) Numerical dominance of small particles vs. biogeochemical significance of large particles, and (3) Eddy-driven variability in particle dynamics and carbon export. This restructuring aims to strengthen the manuscript's overall narrative and facilitate a clearer interpretation of the key findings. Additionally, we have removed the *Data Uncertainty* paragraph and incorporated the relevant content into Section 4.3 of the Discussion to ensure a more integrated and coherent link with the preceding analysis.

The manuscript currently includes an excessive number of figures in the main text. I recommend moving some of these to the Supplementary Information to improve the flow and readability of the paper. In general, the manuscript lacks conciseness and requires significant revision to enhance clarity and focus. Some issues related to formatting and presentation should be addressed. For example, in the Abstract, the abbreviation "ESD" should be spelled out upon first use. Additionally, there is inconsistency in the formatting of parentheses and statistical values (e.g., "mean: xx" vs. "mean = xx"). Additionally, the legends in some figures contain inconsistent font styles or sizes, which affects the overall readability and professionalism of the visuals. A thorough review and standardization of formatting across the manuscript are needed.

**Response:** Thank you for your constructive feedback regarding the clarity and formatting of the manuscript. We fully agree that the number of figures in the original submission was excessive. In the revised version, we have reorganized the figures in accordance with the updated structure and content of the manuscript. The main text now includes 9 figures, and 3 figures in the Supplementary Material. In

addition, we have carefully reviewed and standardized formatting throughout the manuscript. We have spelled out "ESD" (equivalent spherical diameter) upon its first mention in the Abstract. Harmonized the use of statistical expressions (e.g., using "mean: xx" consistently). Standardized font style and size in all figure legends to improve visual consistency and readability. We have also conducted a comprehensive formatting check across the entire manuscript to enhance its overall clarity and professionalism.

Once again, thank you very much for your valuable comments. We sincerely hope that our responses and revisions meet your expectations.

Best regards,

Shujin Guo

---

## Author Response (AR2)

**Dear Editor and Reviewers.**

Thank you very much for your continued time and effort in evaluating our manuscript. We sincerely appreciate the constructive and thoughtful comments provided during the second round of review. Based on your guidance, we have carefully revised the manuscript once again and provide a detailed, point-by-point response to each of the reviewers' comments below. In particular, we have addressed Reviewer #1's concern regarding the absence of diel vertical migration (DVM) discussion, which has now been incorporated into the revised version (Section 4.1). We also followed Reviewer #2's recommendation to streamline Section 4.3 and the Conclusion, and to avoid redundancy and excessive station-level comparisons. Formatting and figure presentation issues were also addressed in detail.

**The following is a detailed account of the specific revisions:**

**For Reviewer #1**

**General comments**

The revised version of your paper answered most of my comments. I especially liked the analysis you added about the nepheloid layer. However, some figures are still to be improved to ease the understanding of the paper. In your answer, you affirmed that the potential effects of DVM were assessed in the new version of the manuscript, however, I did not find anything on that.

Response: Thank you very much for your valuable comments. We have revised the figure colors throughout the manuscript for better clarity and consistency, and we have also provided additional explanations regarding the interpolation artifacts in UVP-derived section plots.

Regarding the potential influence of diel vertical migration (DVM), we assessed particle abundance patterns at the two nighttime sampling stations, S2 and S4. At station S2, a high particle abundance was observed in the bottom layer; however, these particles are overwhelmingly small-sized, with almost no zooplankton detected based on image analysis. Therefore, we believe this high-value layer is unlikely to be related to DVM. In contrast, at station S4, a slight peak in particle abundance was observed in the upper 100 m. Notably, this layer also showed elevated zooplankton abundance (see Supplementary Fig. S2a), suggesting that DVM could have contributed to the elevated particle signal at this station. We have now clarified this point in the revised manuscript (in Discussion 4.1) to address your concern.

**Specific comments**

**Introduction**

I liked the changes you did, especially the details on the eddies and their impact on plankton, and the unanswered questions.

Response: Thank you very much for your positive feedback. These changes were made to better highlight the scientific context and significance of our study, and we are encouraged that you found them helpful. Lines 62/64: this sentence refers to Picheral et al. 2022 which is about the UVP6 and not the UVP5 used in this study. Moreover, it does not seem useful from a scientific point of view.

Response: Thank you for pointing this out. We are correct that Picheral et al. (2022) focuses on the UVP6, while our study utilized the UVP5. We agree that this reference is not directly relevant to our methodology or scientific discussion, and we have removed this sentence from the revised manuscript accordingly.

Line 103: since anticyclonic eddies are more frequently observed in winter and early spring while cyclonic ones are more present during summer, do you know why you have both of them in June? Response: Thank you for your insightful question. Although climatological patterns suggest that cyclonic eddies are more prevalent during summer and anticyclonic eddies during winter and early spring, both types of eddies can co-occur in the same season due to the complex mesoscale dynamics in the South China Sea.

In our case, the two eddies observed during the June cruise were detected based on sea level anomaly. Similar coexistence of cyclonic and anticyclonic eddies in one cruise has also been reported in previous studies (Xiao et al., 2025; Xu et al., 2025), suggesting that their generation mechanisms may include not only seasonal forcing but also factors such as wind stress curl variability (Metzger, 2003; Xiu et al., 2019), current instabilities, and Kuroshio intrusions (Wang et al., 2020). Therefore, it is entirely plausible to observe both cyclonic and anticyclonic eddies during the same month, such as in June, as seen in our study.

Metzger. 2003. Upper ocean sensitivity to wind forcing in the South China Sea. Journal of Oceanography, 59: 783-798.

Wang Q, Zeng L L, Chen J, He Y K, Zhou W D, Wang D X. 2020. The linkage of Kuroshio intrusion and mesoscale eddy variability in the northern South China Sea: subsurface speed maximum. Geophysical Research Letters, 47(11): e20020GL087034.

Xiao Y F, Zhuang Z P, Xin M, Cui T W, Liu R J, Ma Y, Liu H Y, Wang D Q, Xu T F, Chen L. 2025. Bio-optical properties of adjacent cyclonic and anticyclonic eddies in the central and western South China Sea during the summer. Progress in Oceanography, 238: 103547.

Xiu P, Dai M H, Chai F, Zhou K B, Zeng L L, Du C J. 2019. On contributions by wind-induced mixing and eddy pumping to interannual chlorophyll variability during different ENSO phases in the northern South China Sea. Limnology and Oceanography, 64(2): 503-514.

Xu W L, Wang G F, Xing X G, Cornec M, Hayward A, Chen B Z, Chen X H. 2025. Mesoscale eddies drive phytoplankton-mediated biogeochemistry in the South China Sea. Journal of Geophysical Research, 130(6): e2024JG008664.

Material and methods

Line 165: You well precised the UVP version you used, however try to be consistent in the rest of the paper: sometimes you referred to it as UVP, UVP5 or even UVP5-HD.

Response: Thank you for your careful reading and helpful suggestion. We agree that consistency in terminology is important. In the revised manuscript, we have standardized the terminology and now consistency refer to the instrument as UVP5-HD, which accurately reflects the version used in this study. Line 168/170: does this sentence refer to Picheral et al. 2010? Or did you also release some calibration experiments? It is not really clear.

Response: Thank you for pointing this out. The sentence refers to the standard calibration procedure described by Picheral et al. (2010). We did not perform independent calibration experiments in this study, but followed the established pixel-to-metric unit conversion parameters provided and validated by the instrument manufacturer and described in the referenced literature. We have revised the sentence in the manuscript to clarify this point and avoid confusion.

Lines 171: in your answer, you precised the softwares you used. I think it would be better to add them directly in the manuscript too.

Response: Thank you for the suggestion. We have added the names and versions of the image analysis software used in our study directly into the manuscript for clarity and completeness.

Line 211: the formula is now readable, thank you. Did you still used Stokes' Law to determine the sinking velocity of particles? As precised in the first version.

Response: Thank you for the follow-up question. In the revised version, we don't use Stokes' Law to estimate sinking velocity. Instead, we adopted the empirical parameterization proposed by Guidi et al. (2008), which relates particle carbon content and sinking velocity as a function of particle size. This approach allows for direct integration of particle abundance and size spectra into carbon flux estimates, using the formula:  $m(d) \times w(d) = A \cdot d^B$ , with A and B as best-fit parameters derived from global sediment trap and UVP data. This approach has been widely adopted in many UVP-based studies and provides a practical means to estimate carbon flux when in situ sinking velocity measurements are unavailable.

Guidi L, Jackson G A, Stemmann L, Miquel J C, Picheral M, Gorsky G. 2008. Relationship between particle size distribution and flux in the mesopelagic zone. Deep-Sea Research I, 55: 1364-1374. Lines 227/230: Did you used the same methodology for particle abundance? You did not precised it reappeared in the interpolation method.

Response: Thank you pointing this out. Yes, we used the same methodology for particle abundance as for particle volume concentration and POC flux. All abundance data were binned into 5 m vertical intervals and then interpolated along transects to produce section plots. This was omitted in the previous version for brevity, but we have now explicitly clarified it in the revised manuscript to ensure consistency and transparency.

**Results**

Fig. 2: it would have been nice to have the location of the cyclonic and anticyclonic eddy influenced stations as you did in Fig. 3 for example.

Response: Thank you for the suggestion. We have updated Fig. 2 to clearly indicate the stations influenced by cyclonic and anticyclonic eddies, consistent with the presentation in Fig. 3.

Fig. 4/5: why is there some interpolated data at the location of S7 while you said in the caption of Tab. 1 that the instrument was malfunctioning at this station? Especially for particle abundances where data were quite different between S8 and S6. I understand that it is due to the interpolation you used, however, you actually have no idea of the water column at this location, hence I would not be confident in showing any data there.

Response: Thank you for your comment. We fully understand your concern regarding the interpolation across the area between S8 and S6, where UVP data were not available due to instrument malfunction. In the revised manuscript, we indeed attempted to address this by adjusting the interpolation settling in ODV. However, unlikely the interpolated section plots for stratified variables (e.g., Fig. 3), the figures generated from high-resolution 5 m binned UVP data (e.g., Figs. 4, 5, and 7) posed specific challenges.

In these cases, applying stricter gridding constraints (e.g., reducing interpolation radius) resulted in large blank areas not only between stations S8 and S6, but also between other neighboring stations such as S6 and S5 (see figure below), severely affecting the continuity and readability of the section. After careful consideration, we decided to retain the current interpolation settings to preserve visual consistency and clarity across all UVP-derived section plots. That said, we now explicitly note in the figure caption that station S7 had no UVP data and that interpolated values in this region should be interpreted with caution. We hope this clarification addresses your concern.

Lines 321/324: how did you determined the 40% threshold? Only by looking at Fig. 6 or did you do any test?

Response: Thank you for your question. The 40% threshold was used as a visual reference to facilitate interpretation of the vertical profiles, rather than a statistically defined cutoff. It represents a rough division between dominant and subordinate contributions of small particles to PVC, and was marked in Fig. 6 to help illustrate the contrasting patterns between eddy-influenced regions. We have now clarified this point in the figure caption to avoid misunderstanding.

Fig. 6: I guess that the yellow dotdash line represents the 40% threshold you are talking about in the text. Can you precised it in the caption? I think the colours might be confusing, you choose blue for cyclonic eddy influenced stations and red for anticyclonic (in the previous figures) and now red and blue represent large or small particles. Maybe you could try to harmonise colours throughout your entire paper, and if you stick to red and blue for ACE and CE then Fig. 6a could be improved too.

Response: Thank you for your helpful suggestion. We have added an explicit note in the figure caption indicating that the yellow dot-dash line represents the 40% threshold discussed in the main text. To avoid confusion with the color scheme used in previous figures—where red and blue indicated anticyclonic and cyclonic eddy-influenced stations, respectively, we have revised the color scheme in Fig. 6 accordingly. Specifically, in Fig. 6a, stations S4 and S5 (ACE) are now shown in red and pink, while stations S11 and S12 (CE) are displayed in blue and light blue. In panels b-e, we have also adjusted the color palette to eliminate the use of red and blue for particle types, thereby maintaining color consistency throughout the manuscript.

Fig.7: Again, why is there interpolated data at the location of S7? Especially since it is POC flux that you derived from UVP data, that you don't actually have at S7.

Response: Thank you for your observation. As noted in our earlier response, no UVP data were collected at the region between S8 and S6 due to instrument malfunction. We acknowledge that the interpolated fields shown in Fig. 7 include a gap-filling interpolation across this region. During the first round of revision, we attempted to mask or exclude this region entirely (as we did for Fig. 3), but due to the higher vertical resolution (5 m bins) in the UVP-derived datasets, this resulted in visually disruptive gaps in horizontal interpolation fields (as shown in the image provided previously). As a compromise, and to ensure a more coherent and readable figure layout, we decided to retain the interpolated field over the S7 region, while clearly indicating in the caption that no data were actually collected at this station. We hope this approach is acceptable.

Fig.8: Same here for the colours.

Response: Thank you for pointing this out. We have revised the color scheme in Fig. 8b-e to ensure consistency with earlier figures. Specifically, stations influenced by anticyclonic eddies (S4 and S5) are

now shown in shades of red, and those influenced by cyclonic eddies (S11 and S12) in shades of blue, following the color convention used throughout the manuscript.

**Discussion**

Since you now discussed a lot more of the shore influenced stations, I wonder if it would be pertinent to highlight them on the map and figures, as you did for ACE and CE stations.

Response: Thank you for the helpful suggestion. While we agree that shore-influenced stations are relevant to the discussion, we did not specifically highlight them in the figures for two main reasons. First, these stations are already easily distinguishable as the innermost stations on each transect, making additional labeling unnecessary. Second, to maintain figure clarity and avoid excessive categorization, we limited the visual distinction in the plots to the anticyclonic (ACE) and cyclonic (CE) eddy-influenced stations, which are central to the objectives of our study. We believe this approach ensures better visual readability while still allowing readers to identify the nearshore locations if needed.

Lines 447/449: can you elaborate please? One can imagine that the particles at the bottom were a few days before in the top layer? (you give some elements of discussion at the very end of the paragraph talking about zooplankton—maybe it could be nice to have this point just after the highlighted lines) Response: Thank you for your comment. We agree that vertical settling from the surface could be a potential source of deep particle accumulation. However, in our case, the elevated abundance of small particles at the bottom layers of S2 and S10 is unlikely to have originated from surface export. If vertical settling were the dominant process, we would expect a more continuous vertical distribution of particles and similar signals at adjacent stations. However, the particle abundance in the upper water column at S2 and S10 was relatively low, and nearby station S3, which is located close to S2, exhibited much lower particle abundance in the deep layer. This spatial mismatch suggests that the bottom enhancement is unlikely to be due to vertical transport from above. Instead, we propose that the elevated abundance of small particles at S2 and S10 likely originated from lateral or bottom sources. Notably, both stations are located in the mid-slope region, where intermediate nepheloid layers are more likely to occur. Similar features have also been reported in previous studies (Jia et al., 2019; Chen et al., 2024), supporting this interpretation. Additionally, following your suggestion, I moved the discussion of zooplankton to an earlier part of the paragraph.

Jia, Y. G., Tian, Z. C., Shi, X. F., Liu, J. P., Chen, J. X., Liu, X. L., Ye, R. J., Ren, Z. Y., Tian, J. W.: Deep-sea sediment resuspension by internal solitary waves in the northern South China Sea. Sci. Rep., 9, 12137, https://doi.org/10.1038/s41598-019-47886-y, 2019.

Chen, T., Liu, X. L., Bian, C. W., Zhang, S. T., Ji, C. S., Wu, Z. S., Jia, Y. G.: Nepheloid layer structure and variability along the highly energetic continental margin of the northern South China Sea. J. Geophys. Res., 129(2), e2023JC020072, https://doi.org/10.1029/2023JC020072, 2024.

Line 524: Steinberg et al 2023 is only about salp faecal pellets, not zooplankton. Response: Thank you for pointing this out. The reference to Steinberg et al. (2023) has been removed from the revised manuscript to avoid confusion.

Lines 580/588: the paragraph on modelling eddies is quite interesting, however I am not sure it belongs here. The subject seems a little bit far from the primary objectives of the paper, and applying it to a model

doesn't seem that straightforward to me, especially since you don't give any example of models revolving eddies.

Response: Thank you for your comment. We agree that the discussion on biogeochemical modeling was somewhat beyond the primary scope of this study. As suggested, we have removed the related content from the revised version to maintain focus and ensure that the discussion remains closely aligned with our main objectives.

**For Reviewer #2**

Both Discussion section 4.3 and the Conclusion contain redundant content. As background information on eddy-mediated nutrient availability and phytoplankton community structure has already been presented earlier in the manuscript, section 4.3 should focus directly on the observed differences in PVC and particle size composition between the two eddy regions, providing concise, data-driven explanations. Avoid station-by-station comparisons, as the two eddy regions have already been clearly defined. Response: Thank you for your insightful suggestion. We agree that Section 4.3 and the Conclusion previously contained some redundancy, and that the Discussion would benefit from a more focused, data-driven structure. In the revised manuscript, we have streamlined Section 4.3 by removing background information on eddy-induced nutrient dynamics and phytoplankton biomass that had already been introduced earlier. Instead, we now focus directly on the key differences in PVC and size composition between the cyclonic and anticyclonic eddy regions. We also removed the station-by-station comparisons and emphasized broader patterns at the regional scale. The Conclusion section has likewise been revised to avoid repetition and to better synthesize the main findings.

The Conclusion should then succinctly summarize the main findings and highlight their broader implications.

Response: Thank you for this helpful suggestion. In the revised version, we have streamlined the Conclusion to succinctly summarize the key findings of our study and emphasize their broader implications for understanding particle-mediated carbon export in marginal seas. We removed redundant content and focused on presenting a concise synthesis of the main results and their significance. In the Fig. S1 caption, there is an extra space between '\mu' and 'mol' that should be removed. Response: Thank you for pointing this out. We have corrected the unit formatting in the Fig. S1 caption by removing the extra space between '\mu' and 'mol'.

Once again, we sincerely thank you for your constructive feedback. We hope that our revisions and responses have adequately addressed your concerns and meet your expectations.

Best regards, Shujin Guo

---

## Author Response (AR3)

**Dear Editor and Reviewers,**

Thank you very much for your continued time and effort in evaluating our manuscript. We sincerely appreciate the comments provided during the third round of review. We have carefully revised the manuscript. The following is a detailed account of the specific revisions:

**For Reviewer #1**

Thank you for the efforts you made to improve your manuscript, especially the figures. I would just remove the unused acronyms in the abstract (UVP5-HD and ESD) before acceptation. Response: Thank you very much for your positive evaluation and helpful suggestion. We have removed the unused acronyms "UVP5-HD" and "ESD" from the Abstract to improve clarity and readability.

**For Reviewer #2**

I am pleased to recommend this manuscript for publication. Before final acceptance, I suggest ensuring that the colors used to represent the four stations in Fig. 6a and Fig. 8a are consistent, which will improve clarity for readers.

Response: We sincerely appreciate your final comment. In response, we have ensured that the colors representing the four stations Fig. 6a and Fig. 8a are now fully consistent, thereby enhancing visual clarity and coherence across figures.

Once again, we sincerely thank you for your constructive feedback. We hope that our revisions have adequately addressed your concerns and meet your expectations.

Best regards, Shujin Guo